# CNB: A Bayesian Nonparametric Approach to Optimal Conformal Prediction

## Abstract

Conformal Bayes has been shown to yield the optimal (i.e., smallest expected volume) prediction sets among all prediction sets with a $(1 - \alpha)$ coverage guarantee if the model is correctly specified. However, a critical issue arises when the model is misspecified: the resulting prediction sets, while still satisfying the frequentist coverage guarantee, can become inefficient and suboptimal. To address this limitation, we propose a conformal nonparametric Bayes (CNB) prediction approach, an innovative solution that incorporates Bayesian nonparametric procedures within conformal prediction. This hybrid offers significant improvements over existing methods in three key aspects: (i) it retains the strengths of the full conformal Bayes, (ii) the Bayesian nonparametric layer enhances robustness under model uncertainty and induces endogenous clustering in the data, (iii) model complexity adapts to the data. Theoretically, we show that the resulting CNB prediction sets are valid and will converge to the optimal level of efficiency. The proposed CNB prediction approach provides significant improvements over existing methods while ensuring optimality and precise uncertainty quantification.

## 1 Introduction

Constructing prediction sets for the ground truth with highly reliable coverage guarantees is crucial for high-stakes tasks such as autonomous vehicles (Bojarski et al., 2016) and clinical diagnosis (Esteva et al., 2017). Conformal prediction is a widely used distribution-free method for uncertainty quantification that provides prediction sets with reliable coverage guarantees using finite samples without requiring assumptions on the underlying data-generating process beyond exchangeability of the data (Vovk et al., 2005). Formally, for a given sample $Z_{1:n} = \{Y_i, X_i\}_{i=1}^n$, $Z_i \overset{iid}{\sim} \mathbb{P}$ and a coverage level $(1 - \alpha)$, conformal inference constructs a prediction set $C_\alpha(X_{n+1})$ from $Z_{1:n}$ and a new instance $X_{n+1}$ based on existing statistical and machine learning techniques such that

$$\mathbb{P}(Y_{n+1} \in C_\alpha(X_{n+1})) \geq 1 - \alpha. \tag{1}$$

Although a prediction set $C_\alpha(X_{n+1})$ can be derived using conformal inference, concerns remain about the efficiency of $C_\alpha(X_{n+1})$. For instance, a prediction set $C_\alpha(X_{n+1})$ with smaller volume is more informative and, therefore, more efficient. Hoff (2023) shows that full conformal Bayes prediction sets, *under a correctly specified Bayesian model*, achieve optimal efficiency; that is, they yield prediction sets with the minimum expected volume at a given coverage level $(1-\alpha)$. However, when the Bayesian model is misspecified, the resulting full conformal Bayes prediction sets at the $(1-\alpha)$ coverage level may fail to attain volume optimality, limiting the practical utility of conformal inference. Bhagwat et al. (2025) propose conformal Bayesian model averaging (CBMA), which selects from $K$ candidate Bayesian models and constructs conformal sets using Bayesian model averaging (Raftery et al., 1997). The constructed CBMA conformal set is asymptotically optimal if the correct model is among these $K$ candidate models. However, CBMA does not resolve the fundamental issue: there are still questions of how to select those $K$ candidate models and how to select $K$ in the first place. Therefore, a critical open problem is how to identify the *correct* Bayesian model or a sufficiently good approximation to construct optimally efficient conformal prediction sets. This issue has long persisted and remains underexplored in the conformal prediction literature (Hoff, 2023; Fong & Holmes, 2021).

**Contributions:** In this paper, we propose a conformal nonparametric Bayes (CNB) approach to address the challenge of constructing efficient conformal prediction sets under Bayesian model uncertainty. The CNB prediction approach is the first method to construct optimal conformal prediction sets without assuming one or multiple predefined Bayesian model(s). It achieves this task by incorporating a Bayesian nonparametric mixture model within the conformal prediction framework to compute the conformity score, enabling data-driven model selection from a potentially infinite set of candidates. Intuitively, the flexibility of the Bayesian nonparametric mixture model allows it to approximate the true data-generating process, ensuring that the correct model is effectively learned from the data. Theoretically, we show the CNB conformal prediction sets are valid and asymptotically optimal with the required coverage guarantee. In addition, the Bayesian nonparametric mixture model captures endogenous clustering structures in the data, further enhancing the expressiveness and robustness of the prediction sets.

## 2 PRELIMINARIES

We begin by recalling the full conformal inference (Vovk et al., 2005; Lei et al., 2018), the conformal Bayes (Vovk et al., 2005; Wasserman, 2011; Fong & Holmes, 2021), and the Dirichlet process mixture model (Antoniak, 1974; Ghosal & Van der Vaart, 2017).

### 2.1 FULL CONFORMAL INFERENCE

In the full conformal prediction method, a conformity score $\sigma_i$,

$$\sigma_i := \sigma(Z_{1:n+1}; Z_i), \tag{2}$$

that takes $Z_{1:n+1}$ as input and measures how similar a data point $Z_i$ is to the sample $Z_{1:n}$ for $i \in \{1, \cdots, n\}$. The conformity score can be computed using existing statistical or machine learning techniques. A typical example is the negative squared error from a point prediction $-\left(y_i - \hat{f}(x_i)\right)^2$, where $\hat{f}(x)$ is some point predictor fit to the dataset $Z_{1:n+1}$, assumed to be symmetric with respect to permutations of $Z_{1:n+1}$. The key requirement of any conformity score is that it is exchangeable, i.e., $\sigma_i$ is invariant to the permutations of $Z_{1:n+1}$. This requirement is satisfied if $Z_{1:n+1}$ is exchangeable, and the rank of $\sigma_{1:n+1}$ is uniformly distributed among $\{1, \cdots, n+1\}$. Thus, the rank of $\sigma_{n+1}$ is a valid $p-$value. For a new instance with known $X_{n+1}$ and a plug-in value $Y_{n+1} = y$, the rank of $\sigma_{n+1}$ is $r(y) = \frac{1}{n+1} \sum_{i=1}^{n+1} \mathbb{1}(\sigma_i \leq \sigma_{n+1})$. The full conformal prediction set for $(1-\alpha)$ coverage level that satisfies equation 1 is

$$C_\alpha(X_{n+1}) = \{y \in \mathbb{R} : r(y) > \alpha\}. \tag{3}$$

### 2.2 FULL CONFORMAL BAYES

In a Bayesian setting, a natural suggestion for the conformity score is the posterior predictive density,

$$\sigma_i = p(Y_i|X_i, Z_{1:n+1}) = \int f(Y_i|X_i; \theta)\pi(\theta|Z_{1:n+1})d\theta, \tag{4}$$

where $f(y|x; \theta)$ is a likelihood with parameter $\theta \in \Theta$. The posterior of $\theta$ with a prior $\pi(\theta)$ is $\pi(\theta|Z_{1:n+1}) \propto \pi(\theta) \prod_{i=1}^{n+1} f(Y_i|X_i; \theta)$. We can see directly from the posterior that the conformity score in equation 4 is invariant to permutations of $Z_{1:n+1}$, validating the conformal Bayes approach. The conformal prediction set $C_\alpha(X_{n+1})$ based on the conformity score equation 4 is optimal with the smallest expected volume if the Bayesian model is correctly selected (Hoff, 2023).

### 2.3 DIRICHLET PROCESS MIXTURE MODELS

The Dirichlet Process (DP), introduced by Ferguson (1973), is a measure on probability measures. Let $(\mathbb{X}, \mathcal{X})$ be a Polish space. The DP defined on $(\mathbb{X}, \mathcal{X})$ is parametrized by a base measure $H$ and a concentration parameter $a > 0$. A random measure $P$ on $(\mathbb{X}, \mathcal{X})$ is distributed according to a DP, denoted by $P \sim \text{DP}(a, H)$, if for any finite partition $(A_1, \cdots, A_k)$ of $\mathbb{X}$,

$$(P(A_1), \cdots, P(A_k)) \sim \text{Dir}(aH(A_1), \cdots, aH(A_k)),$$

where $\text{Dir}(a_1, \cdots, a_k)$ represents the Dirichlet distribution with parameters $(a_1, \cdots, a_k)$ (see Appendix B). A sample $\theta_{1:n}$ from $\text{DP}(a, H)$ is drawn by

$$\theta_{1:n}|P \overset{iid}{\sim} P; \qquad P \sim \text{DP}(a, H).$$

A more explicit characterization of the DP is the *stick-breaking representation* provided by Sethuraman (1994). If $P \sim \text{DP}(a, H)$, its stick-breaking representation is

$$P = \sum_{i=1}^{\infty} w_i \delta_{\theta_i}, \quad w_i = v_i \prod_{k=1}^{i-1}(1 - v_k), \quad v_i \sim \text{Beta}(1, a). \tag{5}$$

Notably, it is known that $P$ is almost surely discrete from the above stick-breaking representation (Blackwell, 1973; Blackwell & MacQueen, 1973), inducing a natural clustering behavior of the sample $\theta_{1:n}$. The variables $\theta_{1:n}$ are randomly partitioned according to which variables are equal to the same value, with the distribution of the partition obtained from a Pólya urn scheme (Blackwell & MacQueen, 1973). Let $\{\theta_j^*\}_{j=1}^{|\mathbf{c}|}$ denote the distinct values of $\theta_{1:n}$, and $\mathbf{c} = \{c_1, \cdots, c_n\}$ be assignment variables such that $\theta_i = \theta_{c_i}^*$, where $|\mathbf{c}|$ is the number of distinct values in $\mathbf{c}$. The clustering behavior of $\theta_{1:n}$ can be characterized by the urn distribution:

$$\theta_{n+1} = \begin{cases} \theta_j & \text{with probability } \frac{|\mathcal{I}_j|}{n+a} \\ \theta, \theta \sim H & \text{with probability } \frac{a}{n+a} \end{cases}, \tag{6}$$

where $\mathcal{I}_j = \{i : c_i = j\}$, and the cardinality $|\mathcal{I}_j|$ is the number of times the value $\theta_j^*$ occurs in $\theta_{1:n}$.

In the Dirichlet process mixture model (DPMM) (Antoniak, 1974), the DP is used as a nonparametric prior in a hierarchical Bayesian form:

$$X_i|\theta_i \sim f(x_i; \theta_i), \quad \text{for } i = 1, \cdots, n; \quad \theta_i|P \sim P; \qquad P \sim \text{DP}(a, H). \tag{7}$$

Equivalently, based on the stick-breaking representation, DPMM assumes $\{X_i\}$ is generated from a mixture of an infinite number of distributions $f(x)$, i.e.,

$$f(x) = \sum_{i=1}^{\infty} w_i f(x; \theta_i) = \sum_{j=1}^{\infty} w_j^* f(x; \theta_j^*), \tag{8}$$

where $w_j^* = \sum_{i \in \mathcal{I}_j} w_i$, and $\{\theta_i\}$ is a sample from the DP that has distinct values $\{\theta_j^*\}$.

## 2.4 RELATED WORKS

Constructing efficient conformal prediction sets has long been a popular research topic in statistical learning. Among various methods in conformal inference, the full conformal Bayes prediction set (Hoff, 2023) is shown to be optimal in terms of minimal expected length provided that the Bayesian model is correctly specified. Fong & Holmes (2021) developed an algorithm based on the Add-One-In (AOI) strategy for computing full conformal Bayes sets and extended it to partially exchangeable data. To address the issue of model uncertainty, Bhagwat et al. (2025) proposed to construct conformal Bayes prediction sets using Bayesian model averaging over $K$ candidate Bayesian models. Gasparin & Ramdas (2024) introduced a majority vote-based aggregation method on conformal sets and proved that such a merged set suffers a loss in coverage. Similarly, Linusson et al. (2020) proposed an out-of-bag ensemble method within the conformal inference framework, although the validity guarantee is only approximate. Gauraha & Spjuth (2021) proposed a method to merge conformal scores from various models, assuming that each model is trained on distinct datasets. Additionally, conformal inference methods based on aggregated p-values have been developed using data splitting schemes (Vovk, 2015; Carlsson et al., 2014; Linusson et al., 2017; Toccaceli & Gammerman, 2019), but these methods may not be well-calibrated. Overall, despite their innovations, existing methods face several limitations: i) lack of coverage guarantee in the aggregation procedures, ii) no theoretical guarantee of coverage and efficiency, iii) not feasible for full conformal inference with small datasets, iv) inflexibility in adapting model complexity to the data, v) inability to capture the endogenous clustering structure in the data. The proposed CNB prediction approach will address these limitations as introduced in the following sections.

## 2.5 Problem Setup

Let $Z_{1:n} = \{Y_i, X_i\}_{i=1}^n$ be an observed sample, where $Y_i \in \mathbb{R}$ is the outcome and $X_i \in \mathbb{R}^d$ is the feature. The objective of this paper is to construct a conformal prediction set $C_\alpha(X_{n+1})$, which is a set of plausible values of $Y_{n+1}$ given a new feature $X_{n+1}$, based on the sample $Z_{1:n}$. We adopt a Bayesian nonparametric setting and use the posterior predictive density as the conformity score in the conformal prediction procedure.

We aim to address the following critical challenges within the framework.

(i) How can we obtain an optimal conformal prediction set (i.e., one with the smallest expected volume) without knowing the correct Bayesian model?

(ii) How can the model capture the endogenous clustering structure in the data when such structure exists? For example, if the outcome $Y_i$ is the GPA of the $i$-th student with feature $X_i$, then the sample $Z_{1:n}$ may naturally exhibit clustering behavior with respect to (latent) factors such as school, majors, etc..

(iii) The model complexity should adapt to data to enhance the model applicability and to reduce the risks of under/over-fitting.

# 3 CNB: Conformal Nonparametric Bayes

## 3.1 The Construction of CNB Prediction Sets

In real applications, the true data generating process is unknown, making it difficult to identify the correct Bayesian model required for the full conformal Bayes approach with optimality. To address this issue, we replace the conformity score equation 4 in the full conformal Bayes with the posterior predictive density from DPMM, i.e.,

$$\sigma_i^{\text{CNB}} = p(y_i|x_i, Z_{1:n+1}) = \mathbb{E}_{P|Z_{1:n+1}}\left[\int f(y_i|x_i; \theta)dP(\theta)\right], \tag{9}$$

where $P \sim \text{DP}(a, H)$ and the expectation $\mathbb{E}_{P|Z_{1:n+1}}$ is taken with respect to the posterior $P|Z_{1:n+1}$.

The conformity score in equation 9 is valid, since it is a function of the full dataset $Z_{1:n+1}$, and the posterior random probability measure $P|Z_{1:n+1}$ is invariant under permutations of $Z_{1:n+1}$. Therefore, a conformal prediction set, $C_\alpha^{\text{CNB}}(X_{n+1})$, based on the conformity score equation 9 can be constructed through the full conformal steps in Section 2.1. That is,

$$C_\alpha^{\text{CNB}}(X_{n+1}) = \{y \in \mathbb{R} : r(y) > \alpha\}, \tag{10}$$

where $r(y) = \frac{1}{n+1}\sum_{i=1}^{n+1} \mathbb{1}(\sigma_i^{\text{CNB}} \leq \sigma_{n+1}^{\text{CNB}})$.

The benefits of the proposed CNB approach are multi-fold. First, the DPMM based posterior predictive density has been shown (Ghosal et al., 1999; Ghosal & Van Der Vaart, 2001) to converge to the posterior predictive density under the correct Bayesian model. Intuitively, the DPMM procedure will learn the correct Bayesian model from data. That is, even without knowledge of the correct Bayesian model, the CNB prediction set $C_\alpha^{\text{CNB}}(X_{n+1})$ will converge to the optimal full conformal Bayes prediction set as we will state in Theorem 2. Second, the endogenous clustering structure in the data will be captured by the DPMM model as described in Section 2.3. This allows CNB to naturally adapt to latent subgroups that often arise in real datasets. Third, the endogenous clustering behavior induced by the DPMM model is totally data-driven, meaning that the model complexity adapts to the data. Fourth, the proposed CNB framework is flexible in the sense that the DPMM model can be replaced by other Bayesian nonparametric mixture models. Specifically, $P \sim \text{DP}(a, H)$ in equation 9 can be replaced by $P \sim$ Bayesian Nonparametric Priors, the general Bayesian nonparametric priors such as the two-parameter Poisson-Dirichlet process, the normalized random measures with independent increments, the Gibbs-type priors. These general Bayesian nonparametric mixture models will provide valid conformal sets, following the same procedure as in our work, while they provide flexibility on the clustering behavior of the data. For a comprehensive discussion of these Bayesian nonparametric models, we refer readers to numerous references, e.g., Lijoi et al. (2010); De Blasi et al. (2013); Müller & Mitra (2013); Zhang & Hu (2021); Ghosal & Van der Vaart (2017).

## 3.2 THE COMPUTATION OF FULL CNB PREDICTION SETS

In what follows, we will discuss the estimation and computation of $\sigma_i^{\text{CNB}}$ in the full CNB conformal prediction approach. There are two main difficulties: (i) how to compute the conditional expectation in equation 9, (ii) $\sigma_i^{\text{CNB}}$ is computed based on $Z_{1:n+1}$, so we have to train the Bayesian posterior using $\{Z_{1:n}, (y, X_{n+1})\}$ for each grid candidate $y$, which is computationally expensive.

For the difficulty (i), we employ the stick-breaking representation of the DP as introduced in Section 2.3. Therefore,

$$\sigma_i^{\text{CNB}} = p(y_i|x_i, Z_{1:n+1}) = \sum_{j=1}^{\infty} w_j^{(n+1)} f(y_i|x_i; \theta_j^{(n+1)}) , \qquad (11)$$

where $w_j^{(n+1)}$ is the posterior weights of $w_j$ updated with $Z_{1:n+1}$, and $\theta_j^{(n+1)}$ is the posterior parameters via posterior inference.

We then truncate the infinite sum in equation 11 at some $K_n$ such that $1 - \sum_{j=1}^{K_n} w_j^{(n+1)} < \epsilon$, where $\epsilon > 0$ is some pre-determined error. Therefore, $\sigma_i^{\text{CNB}} \approx \sum_{j=1}^{K_n} w_j^{(n+1)} f(y_i|x_i; \theta_j^{(n+1)})$.

For the difficulty (ii), we use the Add-One-In (AOI) importance sampling strategy introduced in Fong & Holmes (2021) as follows. We obtain the posterior sampling $\{w_{j,t}^{(n)}, \theta_{j,t}^{(n)}\}_{j=1,t=1}^{K_n,T}$ conditional only on $Z_{1:n}$ using MCMC (for example, Algorithm 8 in Neal (2000) and Chapter 5 in Ghosal & Van der Vaart (2017)) with $T$ iterations. For a specific candidate point $(Y_{n+1}, X_{n+1}) = (y, x)$, the AOI trick approximates the updated posterior using importance weights

$$\widetilde{w}_{j,t}^{(n)} = \frac{w_{j,t}^{(n)} f(y|x; \theta_{j,t}^{(n)})}{\sum_{k=1}^{K_n} w_{k,t}^{(n)} f(y|x; \theta_{k,t}^{(n)})} . \qquad (12)$$

The conformity score in equation 11 for any $(Y_i, X_i)$ can be eventually computed using

$$\widehat{\sigma}_i^{\text{CNB}} := \widehat{\sigma}^{\text{CNB}}(Y_i, X_i) = \frac{1}{T} \sum_{t=1}^{T} \sum_{j=1}^{K_n} \widetilde{w}_{j,t}^{(n)} f(Y_i|X_i; \theta_{j,t}^{(n)}) . \qquad (13)$$

Let $\mathcal{Y}$ be a grid of candidate $y$ values. Then, $C_\alpha^{\text{CNB}}(X_{n+1})$ can be computed by

$$\widehat{C}_\alpha^{\text{CNB}}(X_{n+1}) = \{y \in \mathcal{Y} : \widehat{r}(y) > \alpha\} , \qquad (14)$$

where $\widehat{r}(y) = \frac{1}{n+1} \sum_{i=1}^{n+1} \mathbb{1}(\widehat{\sigma}_i^{\text{CNB}} \leq \widehat{\sigma}_{n+1}^{\text{CNB}})$. We summarize the whole steps of obtaining equation 14 in Algorithm 1.

---

**Algorithm 1** CNB: Conformal Nonparametric Bayes prediction algorithm

---

**Input:** Sample $Z_{1:n}$ and the test point $X_{n+1}$. A predefined miscoverage level $\alpha$ and a tolerance error $\epsilon$.
**Output:** The prediction set $C_\alpha^{\text{CNB}}(X_{n+1})$.
**Obtain:** Posterior samples $\{w_{j,t}^{(n)}, \theta_{j,t}^{(n)}\}_{j=1,t=1}^{K_n,T}$ conditional on $Z_{1:n}$ of the DPMM model using MCMC such that $\sum_{j=1}^{K_n} w_{j,t}^{(n)} \geq 1 - \epsilon$.
**For** $y \in \mathcal{Y}$: Compute $\widehat{\sigma}^{\text{CNB}}(X_{n+1}, y)$ using equation 13 and compute $\widehat{r}(y)$.
**End For**
**Return:** Report $\widehat{C}_\alpha^{\text{CNB}}(X_{n+1}) = \{y \in \mathcal{Y} : \widehat{r}(y) > \alpha\}$.

---

## 3.3 SPLIT CNB CONFORMAL GENERALIZATION

Split conformal inference (Papadopoulos et al., 2002; Lei et al., 2018) is commonly used to mitigate the high computational complexity of full conformal inference. In the split conformal approach, data are divided into a training dataset $\mathcal{I}_{\text{train}}$ and a calibration dataset $\mathcal{I}_{\text{cal}}$. The conformity score is trained on $\mathcal{I}_{\text{train}}$, while the empirical $1 - \alpha$ quantile is computed using $\mathcal{I}_{\text{cal}}$. Although the split conformal

approach does not use all the data as effectively as full conformal inference, it significantly simplifies the computation.

The proposed CNB conformal prediction method can be naturally generalized to a split CNB conformal prediction. Specifically, the CNB conformity score equation 11 can be directly used within the classic split conformal prediction framework. When both the training size $n_{\text{train}}$ and the calibration size $n_{\text{cal}}$ tend to infinity, the split CNB prediction set is asymptotically optimal.

### 3.4 The Choices of $f(y|x;\theta)$

For real applications, we need to identify $f(y|x;\theta)$ to compute the CNB conformity score as introduced in this section. There are two typical choices.

**1. DPMM with joint Gaussian modeling.**

The joint distribution of $(x,y) \in \mathbb{R}^{d+1}$ follows a Gaussian distribution, e.g., $(x,y)|\theta \sim \mathcal{N}(\mu,\Sigma)$, where $(\mu,\Sigma) = \left( \begin{bmatrix} \mu_x \\ \mu_y \end{bmatrix}, \begin{bmatrix} \Sigma_{xx} & \Sigma_{xy} \\ \Sigma_{yx} & \Sigma_{yy} \end{bmatrix} \right)$. Therefore, the posterior predictive is

$$f(y|x,\theta) = \mathcal{N}(y|\mu_{y|x}, \Sigma_{y|x}),  \tag{15}$$

where $\mu_{y|x} = \mu_y + \Sigma_{yx}\Sigma_{xx}^{-1}(x - \mu_x)$ and $\Sigma_{y|x} = \Sigma_{yy} - \Sigma_{yx}\Sigma_{xx}^{-1}\Sigma_{xy}$. Normally, a Normal-Inverse-Wishart (NIW) prior is placed on the full joint parameters, e.g., $(\mu,\Sigma) \sim \text{NIW}(\mu_0, \kappa_0, \Psi_0, \nu_0)$ with some hyperparameters $(\mu_0, \kappa_0, \Psi_0, \nu_0)$, to complete a full Bayesian procedure.

**2. DPMM with regression models.** We directly assume the predictive function

$$f(y|x,\theta) = \mathcal{N}(g(x;\beta), \Sigma_y),  \tag{16}$$

where $\Sigma_y \sim \text{InverseGamma}(a_0, b_0)$ with hyperparameters $(a_0, b_0)$ and $g(x,\beta)$ can be used to specify the linear or nonlinear dependence between $x$ and $y$. For example,

1. A linear regression, $g(x,\beta) = x^\top \beta$ with priors $\beta \sim \mathcal{N}(0,\tau)$ and hyperparameter $\tau$.
2. A polynomial regression, $g(x,\beta) = \phi(x)^\top \beta$, where $\phi(x) = (1, x, \cdots, x^d) \in \mathbb{R}^{d+1}$ and $\beta \sim \mathcal{N}(0,\tau)$.
3. A Gaussian process (GP) based regression (Williams & Rasmussen, 1995; Schulz et al., 2018), $g(x,\beta) = \text{GP}(\psi(x), \kappa(x,x')) + e$, $e \sim \mathcal{N}(0, \Sigma_y)$, where $\psi(x) \sim \text{GP}(0, \kappa(x,x'))$ and $\kappa(x,x')$ is some kernel, e.g., the Radial Basis Function (RBF) kernel $\kappa(x,x') = \rho^2 \exp\{-||x - x'||^2/(2\ell^2)\}$.
4. A neural network (NN) based regression, $g(x,\beta) = \text{NN}(x;\beta)$, where $\beta$ denotes the neural network weights (could have a prior $\beta \sim \mathcal{N}(0,\tau)$) updated by backpropagation.

## 4 Theoretical Results of CNB

In this section, we show the validation of the proposed CNB prediction set, the theoretical guarantee of its optimality and the induced clustering behavior.

**Theorem 1 (Validation and coverage guarantee of CNB)** *The proposed CNB conformity score in equation 9 (or equation 11) is invariant to permutations of $Z_{1:n+1}$, thus is valid. Therefore,*

$$\mathbb{P}(Y_{n+1} \in C_\alpha^{CNB}(X_{n+1})) \geq 1 - \alpha,  \tag{17}$$

*where $C_\alpha^{CNB}(X_{n+1})$ is given in equation 10. Moreover, if $\sigma_{1:n}^{CNB}$ are almost surely distinct, we have*

$$1 - \alpha \leq \mathbb{P}(Y_{n+1} \in C_\alpha^{CNB}(X_{n+1})) \leq 1 - \alpha + \frac{1}{n+1}.  \tag{18}$$

The key to validating $\sigma_i^{\text{CNB}}$ is that the posterior random probability measure $P|Z_{1:n+1}$ is invariant under permutations of $Z_{1:n+1}$. The coverage guarantees equation 17 and equation 18 of the proposed CNB prediction set follow directly from the classic conformal inference (Vovk et al., 2005; Lei et al., 2018). Theorem 1 provides the foundation of the CNB conformal prediction set. We will further show in the next result that the proposed CNB conformal prediction set is asymptotically optimal in that it achieves the minimum expected volume among all valid conformal prediction sets. As a result, the CNB prediction set is most informative asymptotically.

**Theorem 2 (Asymptotic optimality of CNB)** *Let $\sigma_i^*$ and $C_\alpha^*(X_{n+1})$ be the conformity score and the full conformal Bayes prediction set with true Bayesian model.* For any prediction set $C$, let $\mathcal{V}(C)$ be the length of the set. *Under the mild assumptions given in Theorem 2 of Tokdar (2006), we have $\sigma_i^{CNB} \to \sigma_i^*$ in probability as $n \to \infty$ ($n_{train} \to \infty$ in the split CNB setting). The convergence is almost surely under some further assumptions given in Ghosal et al. (1999). Furthermore, $C_\alpha^{CNB}(X_{n+1})$ is asymptotically optimal, i.e., $\lim_{n\to\infty} |\mathbb{E}[\mathcal{V}(C_\alpha^{CNB}(X_{n+1}))] - \mathbb{E}[\mathcal{V}(C_\alpha^*(X_{n+1}))]| = 0$ ($n_{train}, n_{cal} \to \infty$ in the split CNB setting).*

Theorem 2 highlights the strong practical applicability of the proposed CNB prediction set. Specifically, it demonstrates that the CNB prediction set is highly informative and efficient, achieving the smallest expected volume asymptotically among all valid conformal sets. This makes CNB particularly suitable for high-stakes prediction tasks where both reliable coverage and efficiency are crucial. The assumptions in Theorem 2 of Tokdar (2006) include (i) $p(y|x, Z_{1:n+1})$ has compact support or light tails, (ii) the true joint distribution $p^*(y, x)$ lies within the Kullback–Leibler support of the DPMM prior, i.e., $p^*(y, x) \in \{p^* : \forall \epsilon > 0, \Pi(p : D_{KL}(p^*||p) < \epsilon) > 0\}$, where $\Pi$ represents the DPMM prior and $D_{KL}$ is the Kullback–Leibler divergence. All these assumptions are satisfied in our setting under the DPMM prior with choices of $f(y|x; \theta)$ as described in Section 3.4. In the next result, we will explore the distribution of the clustering behavior induced by the DPMM prior.

**Theorem 3 (The induced clustering behavior of CNB)** *In the CNB framework, $(\theta_i = \theta_j)$ is possible with positive probability, $Z_{1:n+1}$ are thus clustered accordingly. Let $\mathbf{c} = \{c_1, \cdots, c_n\}$ be assignment variables such that $\theta_i = \theta_{c_i}^*$, where $|\mathbf{c}|$ is the number of distinct values in $\mathbf{c}$. Then,*

$$\mathbb{P}(c_1, \cdots, c_n) = \frac{a^{|\mathbf{c}|} \prod_{j=1}^{|\mathbf{c}|}(n_j - 1)!}{a(a+1)\cdots(a+n-1)}, \tag{19}$$

$$\mathbb{P}(c_1, \cdots, c_n | Z_{1:n}) \propto \left( \prod_{j=1}^{|\mathbf{c}|} a \prod_{i:c_i=j} f(y_i|x_i; \theta_j^*) \right) \prod_{j=1}^{|\mathbf{c}|} \int \prod_{i:c_i=j} f(y_i|x_i; \theta_j^*) dH(\theta_j^*), \tag{20}$$

*where $n_j$ is the number of cluster $j$. For the clustering number $|\mathbf{c}|$, we have*

$$\mathbb{E}[|\mathbf{c}|] = \sum_{i=1}^{n} \frac{a}{a+i-1}, \qquad \lim_{n\to\infty} \frac{|\mathbf{c}|}{\log n} = a \ a.s.. \tag{21}$$

The results in Theorem 3 follow directly from the results in Antoniak (1974); Pitman (2006); Aldous (2006); Teh et al. (2006). Theorem 3 characterizes the distribution of the clustering and the clustering number induced by the CNB method, providing the intuitions of how the clustering behaves. For instance, the clustering number $|\mathbf{c}|$ is increasing in $n$ with a slower rate $\log n$, which characterizes the model complexity of the DPMM. Importantly, the induced clustering behavior is totally data-driven.

## 5 EXPERIMENTS

To clearly demonstrate the performance of the proposed CNB prediction method, we compare the average lengths of the prediction sets of CNB and the split CNB (CNB-split) with two conformal Bayesian methods: CBMA (Bhagwat et al., 2025) and full conformal Bayes (CB) (Hoff, 2023). We also compare our methods with three non-Bayesian conformal methods: Conformalized Quantile Regression (CQR) (Romano et al., 2019), Localized Conformal Prediction (LCP) (Guan, 2023), Conformal Prediction with Conditional Guarantees (CPCG) (Gibbs et al., 2025).

We run these methods on one simulated dataset from a mixture of 3 Gaussian components. With known ground true data generation distribution of the simulated dataset, we are able to specify the "correct" Bayesian model, under which, we obtain the full conformal Bayesian prediction sets as the "optimal" baseline. To emphasize the applicability of the proposed CNB method, we make comparisons on four real datasets from UCI (Dua et al., 2017): California housing dataset, community crime dataset, bike sharing demand dataset, and concrete compressive strength dataset.

## 5.1 SIMULATION STUDY

In this simulation study, we simulate a dataset from a mixture of 3 Gaussian components as follows.

$$Y|X, C = k \sim \mathcal{N}(\mu_k(X), \sigma^2), \quad X = (X_1, X_2), \quad \mu_1(X) = a_0 + a_1 X_1 + a_2 X_2^2,$$
$$\mu_2(X) = b_0 + b_1 X_1^2 + b_2 X_1 X_2, \quad \mu_3(X) = c_0 + c_1 X_1^2 + c_2 \cos(X_2),$$

where $X_1 \sim \text{Unif}(-1, 2)$, $X_2 \sim \mathcal{N}(1, 0.5)$, $\sigma^2 = 0.5$. $Y$ is randomly drawn from one of the three components by $\mathbb{P}(C = k) = \frac{1}{3}$ for each $k \in \{1, 2, 3\}$. The coefficients are also randomly drawn from different uniform or Gaussian distributions (given in Appendix C). With the known data generation distribution, we include three different Bayesian models for both CB and CBMA. i) CB1: CB with the true Bayesian model, ii) CB2: CB with partially true Bayesian model (three components include two true components), iii) CB3: CB with misspecified Bayesian model, iv) CBMA1: CBMA with the true Bayesian model as one candidate model, v) CBMA2: CBMA with one partially true Bayesian model as one candidate model, vi) CBMA3: CBMA with misspecified Bayesian candidate models.

We derive the conformal prediction sets for the 7 models when the sample size $n$ takes values in $\{100, 300, 600, 1000\}$. The mean lengths of the prediction sets at level $\alpha = 0.2$ are shown in Figure 5.1. When $n = 1000$, the mean interval lengths of prediction sets for CNB, CNB-split, CBMA1, CB1 are $3.233, 3.237, 3.463, 3.303$, respectively. According to the theoretical results of Hoff (2023), the prediction set of CB1 is asymptotically optimal, therefore, CB1 at $n = 1000$ can be viewed as the "optimal" baseline. CBMA1, which includes the true Bayesian model among the candidate models, also shows good performance. However, CBMA2 and CBMA3, which do not include the true Bayesian model, do not guarantee optimality. The proposed CNB prediction sets are asymptotically optimal as claimed in Theorem 2, despite not incorporating any knowledge of the true model. The split CNB prediction set is also optimal when $n$ is large, but it is less competitive when $n \leq 300$ as expected. Moreover, CNB prediction sets consistently have shorter lengths than the predictions sets from CBMA, highlighting the sufficiency the CNB approach. In addition, we can see the lengths of the split CNB prediction sets decrease rapidly when the sample size increases. Therefore, the split CNB approach is preferable when computational efficiency is a priority while still achieving competitive results. Both the non-Bayesian methods do not show better results than those from CNB, CB and CBMA in this case. The coverage rates for those models are closely around $0.8$, they are summarized in Table 1 in Appendix C.

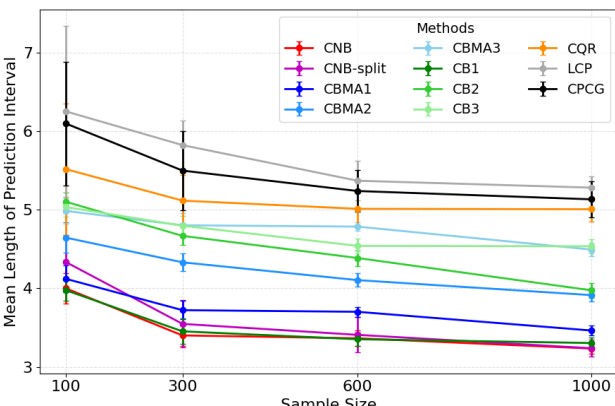

Figure 1: Simulation Study: Comparison of the mean length of the prediction sets from CNB, CNB-split, CBMA1 (include the true Bayesian model), CBMA2, CBMA3, CB1 (with the true Bayesian model), CB2, CB3, CQR, LCP, CPCG with respect to different sample sizes.

## 5.2 REAL DATA EXPERIMENT

We now evaluate the performance of the proposed CNB methods and compare the results with those from other methods on four real datasets. In particular, for the California housing dataset, we consider three different Bayesian models for CB and CBMA based on our understanding of the

dataset. Details of the datasets and the specific models are provided in Appendix D. The mean lengths of the prediction sets from all models are given in Figure 5.2. The corresponding coverage rates for these models are around $0.8$, they are reported in Appendix D. The proposed CNB method has competitive performance comparing with other methods, even when $n$ is small. Although LCP and CPCG also give very narrow prediction sets (e.g., on Concrete, Crime, Bike), CNB consistently give the optimal prediction sets, demonstrating its applicability on those datasets.

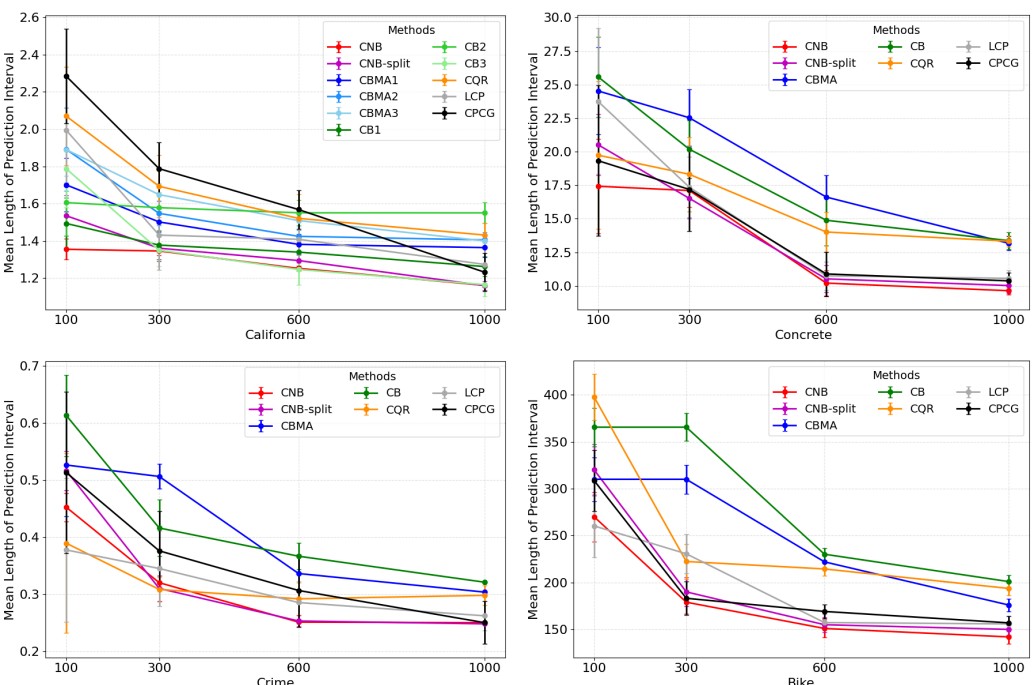

Figure 2: Real Data: Comparison of the mean length of the prediction sets from CNB, CNB-split, CBMA, CB, CQR, LCP and CPCG with respect to different sample sizes on four datasets.

## 6 DISCUSSION

In this work, we address a fundamental challenge in traditional conformal inference: How to construct a prediction set with the smallest length. It is known (Hoff, 2023) the full conformal Bayesian prediction set with a correct specified Bayesian model can produce such optimal sets, however, it is challenging to have such a correct specified model.

To overcome this, we propose the CNB (Conformal Nonparametric Bayes) prediction approach, which incorporates a Bayesian nonparametric procedure in the construction of the conformity score. This allows CNB to produce asymptotically optimal conformal prediction sets, even in the absence of prior model information. Moreover, the nonparametric Bayesian method naturally captures the endogenous clustering structure in the data, thereby enhancing the accuracy of the conformity score. In computation, i) we present an effective algorithm for constructing a full CNB prediction set, ii) the CNB conformity score can be naturally used in split conformal inference.

However, several limitations remain, which point to directions for future research. First, the posterior sampling method used in Bayesian nonparametric inference, such as MCMC, Gibbs sampling, or variational Bayes, can influence the quality and sufficiency of the conformity score. Second, in our implementation of full CNB prediction, we rely on the AOI approximation to avoid refitting the model for each candidate value on the prediction grid. This approximation may suffer from high variance, as discussed in Fong & Holmes (2021). A practical alternative, particularly when the sample size is sufficiently large ($n \geq 300$ in our experiments), is to use the CNB-split method, which offers a more stable and computationally efficient solution.

ETHICS STATEMENT

All authors have read and commit to adhering to the ICLR Code of Ethics. We are not aware of any ethical concerns, and this work does not violate the ICLR Code of Ethics.

REPRODUCIBILITY STATEMENT

We facilitate reproduction of all results. The simulation experiment setup, and implementation details needed to replicate our experiment in Section 5.1 are described in Appendix C. The data generation process is included in our code scripts in the supplementary materials. For the real data experiment in Section 5.2, we use four widely adopted, open-access datasets from UCI, namely California Housing dataset, community crime dataset, bike sharing demand dataset, and concrete compressive strength dataset. Model specifications are provided in Appendix D. Complete proofs of all theoretical results, together with the assumptions under which they hold, appear in Appendix E. An anonymized code repository containing scripts for reproducing the results in tables and figures is provided in the supplemental materials.

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

## A  STATEMENT OF USING LLM

We used ChatGPT solely to proofread the Introduction section (Section 1), limited to grammar checking and spelling.

## B  ADDITIONAL DEFINITIONS AND DISCUSSIONS

In this section, we include some extra definitions and discussion of the proposed CNB method.

### B.1  DEFINITIONS AND EXTRA RESULTS

**Definition 4** *A random vector* $(X_1, \cdots, X_k) \sim Dir(a_1, \cdots, a_k)$ *with parameters* $a_i > 0$ *for all* $i$, *has the density function as follows.*

$$f(x_1, \cdots, x_k) = \frac{\Gamma(\sum_{i=1}^k a_i)}{\prod_{i=1}^k \Gamma(a_i)} \prod_{i=1}^k x_i^{a_i - 1},$$

*where* $\sum_{i=1}^k x_i = 1$ *and* $x_i \in [0, 1]$ *for all* $i$.

In Section 3.2, the computation of full CNB prediction set, we truncate the infinite sum in equation 11 at some $K_n$ such that $1 - \sum_{j=1}^{K_n} w_j^{(n+1)} \le \epsilon$. This truncation does not break the exchangeability of the corresponding conformity scores.

**Remark 5** *The truncation* $K_n$ *is a deterministic function of the posterior weights* $\{w_j^{(n+1)}\}$, *hence a symmetric functional of the set* $Z_{1:n+1}$. *The posterior sequence* $\{(w_j^{(n+1)}, \theta_j^{(n+1)})\}$ *is unchanged by permuting* $Z_{1:n+1}$. *Therefore, the truncation index* $K_n$ *remains the same before and after the permutation. The truncation* $K_n$ *does not break the exchangeability of the corresponding conformity scores.*

We apply the AOI trick in the computation the CNB conformity scores to avoid re-training of the model and reduce the computational cost. If we replace $C_\alpha^{CNB}(X_{n+1})$ by its AOI approximation $\widehat{C}_\alpha^{CNB}(X_{n+1})$, the approximation error will affect the coverage rate in Theorem 1.

**Remark 6** *Assume that there exists* $\epsilon > 0$ *and* $\delta \in [0, 1)$ *such that*

$$\mathbb{P}\left(\sup_{y \in \mathcal{Y}} \max_{1 \le i \le n+1} |\widehat{\sigma}_i^{CNB} - \sigma_i^{CNB}| \le \epsilon\right) \ge 1 - \delta. \tag{22}$$

*We further assume the density of* $\sigma_{n+1}^{CNB}$ *is bounded by* $L$. *Then, we have*

$$\mathbb{P}\left(Y_{n+1} \in \widehat{C}_\alpha^{CNB}(X_{n+1})\right) \ge 1 - \alpha - \delta - 4L\epsilon. \tag{23}$$

**Proof:**

For any $y \in \mathcal{Y}$, we recall that $\widehat{\sigma}_i^{CNB}$ and $\sigma_i^{CNB}$ are evaluated on $\{(X_i, Y_i)\}_{i=1}^n \cup \{(X_{n+1}, y)\}$. For each $i \in \{1, \cdots, n+1\}$, if $|\widehat{\sigma}_i^{CNB} - \sigma_i^{CNB}| \le \epsilon$, then a label flip in the indicator $\mathbb{1}(\widehat{\sigma}_i^{CNB} \le \widehat{\sigma}_{n+1}^{CNB})$ relative to $\mathbb{1}(\sigma_i^{CNB} \le \sigma_{n+1}^{CNB})$ can occur only if $|\sigma_i^{CNB} - \sigma_{n+1}^{CNB}| \le 2\epsilon$. Therefore,

$$|\widehat{r}(y) - r(y)| \le \frac{1}{n+1} \sum_{i=1}^{n+1} \mathbb{1}(|\sigma_i^{CNB} - \sigma_{n+1}^{CNB}| \le 2\epsilon). \tag{24}$$

We denote $\Gamma_n(y) := \frac{1}{n+1} \sum_{i=1}^{n+1} \mathbb{1}(|\sigma_i^{CNB} - \sigma_{n+1}^{CNB}| \le 2\epsilon)$ for notational simplicity. For $y = Y_{n+1}$, we have $|\widehat{r}(Y_{n+1}) - r(Y_{n+1})| \le \Gamma_n(Y_{n+1})$.

Under the conformal setting with a continuous score, the rank $r(Y_{n+1})$ is uniformly distributed on $\{\frac{1}{n+1}, \cdots, 1\}$, then, for any $t \in [0, 1]$, $\mathbb{P}(r(Y_{n+1}) \le t) \le t$. Hence,

$$\mathbb{P}(r(Y_{n+1}) \le \alpha + \Gamma_n(Y_{n+1})) = \mathbb{E}[\mathbb{P}(r(Y_{n+1}) \le \alpha + \Gamma_n(Y_{n+1})|\Gamma_n(Y_{n+1}))]$$
$$\le \mathbb{E}[\alpha + \Gamma_n(Y_{n+1})] = \alpha + \mathbb{E}[\Gamma_n(Y_{n+1})].$$

Therefore, on the event $E := \sup_{y \in \mathcal{Y}} \max_{1 \leq i \leq n+1} |\widehat{\sigma}_i^{CNB} - \sigma_i^{CNB}| \leq \epsilon$,

$$\mathbb{P}(Y_{n+1} \notin \widehat{C}_\alpha^{CNB}(X_{n+1})) \leq \mathbb{P}(r(Y_{n+1}) \leq \alpha + \Gamma_n(Y_{n+1})) \leq \alpha + \mathbb{E}[\Gamma_n(Y_{n+1})].$$

Under the assumption $\mathbb{P}(E^c) \leq \delta$, we have $\mathbb{P}(Y_{n+1} \notin \widehat{C}_\alpha^{CNB}(X_{n+1})) \leq \alpha + \mathbb{E}[\Gamma_n(Y_{n+1})] + \delta$.

Now, by the definition of $\Gamma_n(Y_{n+1})$, we have

$$\mathbb{E}[\Gamma_n(Y_{n+1})] = \mathbb{P}(|\rho - \rho'| \leq 2\epsilon),$$

where $\rho, \rho'$ are iid copies of $\sigma_{n+1}^{CNB}$ evaluated on $Z_{1:n+1}$. Due to the fact that $f_{\rho-\rho'}(v) = \int f_\rho(u) f_\rho(u - v) dv$, that satisfies $\|f_{\rho-\rho'}\|_\infty \leq \|f_\rho\|_\infty \leq L$ by assumption. Thus,

$$\mathbb{E}[\Gamma_n(Y_{n+1})] = \mathbb{P}(|\rho - \rho'| \leq 2\epsilon) = \int_{-2\epsilon}^{2\epsilon} f_{\rho-\rho'}(v) dv \leq 4\epsilon \|f_\rho\|_\infty \leq 4L\epsilon.$$

Therefore,

$$\mathbb{P}(Y_{n+1} \notin \widehat{C}_\alpha^{CNB}(X_{n+1})) \leq \alpha + \mathbb{E}[\Gamma_n(Y_{n+1})] + \delta \leq \alpha + \delta + 4L\epsilon.$$

This completes the proof.

## B.2 Discussion of CNB based on upstream models

CNB is model agnostic and composes naturally with learned representations. Let $g : \mathbb{R}^d \to \mathbb{R}^s$ be any upstream model (e.g., a tabular or time series backbone) trained on the training split only, and define an embedding $\phi(x) = g(x)$. We may (i) run representation based CNB, replacing $x$ by $\phi(x)$ and fitting the BNP conditional $p(y|\phi(x))$(e.g., a DPMM for $(\phi(x), y)$), or (ii) run a residual CNB, using a backbone baseline $\mu_g(x)$ (and optionally $\sigma_g^2(x)$) and modeling the residual $r = y - \mu_g(x)$ with a BNP mixture so that the predictive for $y|x$ is the convolution of the baseline and the BNP residual model.

Both cases only change the predictive distributions used in the computation of the CNB scores and leave the conformal machinery untouched. Since $\phi$ (and $\mu_g(x)$) are fixed prior to calibration, exchangeability is preserved and the finite sample coverage guarantee of Theorem 1 remains valid. In practice, using $\phi(x)$ reduces effective dimension and let the BNP approach discover local regimes in representation space, often yielding shorter sets at the same coverage. This makes CNB a representation compatible conformal method, practitioners can keep their preferred pretrained model and simply use the outputs into the CNB predictive.

## C Settings and Additional Results of Simulation Study

All the experiments are done on a MacBook Pro with Apple M3 Pro chip and $18$ GB memory. Given the posterior samples of the model, the computational complexity of obtaining the conformal set for each $X_{n+1}$ is $O(nTn_{\text{grid}})$ (Fong & Holmes, 2021), where $n_{\text{grid}}$ is the cardinality of $\mathcal{Y}$. In Table 3, we report the mean computation time for each method on the simulation study.

### C.1 Setting Details of Simulation Study

The coefficients $\{a_0, a_1, a_2, b_0, b_1, b_2, c_0, c_1, c_2\}$ are randomly drawn from either uniform distribution or Gaussian distribution with some randomly generated parameters. Specifically, $a_0 \sim \text{Unif}(-4, -2)$, $a_1 \sim \text{Unif}(0.8, 1.5)$, $a_2 \sim \mathcal{N}(0.5, 1)$, $b_0 \sim \text{Unif}(-0.5, 0.6)$, $b_1 \sim \text{Unif}(0.5, 1.2)$, $b_2 \sim \text{Unif}(0.4, 1)$, $c_0 \sim \mathcal{N}(2.2, 0.9)$, $c_1 \sim \text{Unif}(-1, -0.3)$, $c_2 \sim \text{Unif}(0.8, 1.5)$.

The coverage rates of the simulation experiments are given as follows in Table 1.

For CNB prediction method, the concentration parameter is learned in Bayesian ways with a prior Gamma(1,1).

We present the average clustering numbers of the CNB approach with respect to different sample sizes in Table 2.

Table 1: Simulation Study: Comparison of the coverage rate for CNB, CNB-split, CBMA1 (include the true Bayesian model), CBMA2, CBMA3, CB1 (with the true Bayesian model), CB2, CB3, CQR, LCP, CPCG with respect to different sample sizes.

| $n$ | CNB | CNB-split | CBMA1 | CBMA2 |
|---|---|---|---|---|
| 100 | $0.796 \pm 0.110$ | $0.799 \pm 0.052$ | $0.802 \pm 0.018$ | $0.790 \pm 0.018$ |
| 300 | $0.813 \pm 0.067$ | $0.801 \pm 0.039$ | $0.810 \pm 0.018$ | $0.797 \pm 0.018$ |
| 600 | $0.810 \pm 0.059$ | $0.792 \pm 0.040$ | $0.798 \pm 0.018$ | $0.799 \pm 0.018$ |
| 1000 | $0.803 \pm 0.027$ | $0.796 \pm 0.024$ | $0.801 \pm 0.018$ | $0.802 \pm 0.018$ |

| $n$ | CBMA3 | CB1 | CB2 | CB3 |
|---|---|---|---|---|
| 100 | $0.797 \pm 0.018$ | $0.794 \pm 0.018$ | $0.810 \pm 0.018$ | $0.781 \pm 0.019$ |
| 300 | $0.800 \pm 0.018$ | $0.820 \pm 0.017$ | $0.797 \pm 0.017$ | $0.793 \pm 0.018$ |
| 600 | $0.807 \pm 0.016$ | $0.810 \pm 0.016$ | $0.798 \pm 0.018$ | $0.799 \pm 0.018$ |
| 1000 | $0.802 \pm 0.018$ | $0.799 \pm 0.016$ | $0.793 \pm 0.017$ | $0.796 \pm 0.018$ |

| $n$ | CQR | LCP | CPCG |
|---|---|---|---|
| 100 | $0.812 \pm 0.082$ | $0.807 \pm 0.078$ | $0.811 \pm 0.085$ |
| 300 | $0.789 \pm 0.032$ | $0.806 \pm 0.031$ | $0.802 \pm 0.040$ |
| 600 | $0.793 \pm 0.026$ | $0.800 \pm 0.028$ | $0.801 \pm 0.028$ |
| 1000 | $0.803 \pm 0.020$ | $0.797 \pm 0.022$ | $0.804 \pm 0.026$ |

Table 2: Simulation Study: Clustering numbers of the CNB approach.

| Clustering Number | 100 | 300 | 600 | 1000 |
|---|---|---|---|---|
| CNB | 6 | 7 | 7 | 8 |
| CNB-split | 17 | 19 | 20 | 20 |

# D  SETTING DETAILS OF REAL DATA EXPERIMENT

## D.1  CALIFORNIA HOUSING DATASET

The dataset describes the housing values across California districts from the 1990 U.S. census. It contains 20640 data points, and the response variable is the median house value, expressed in hundreds of thousands of dollars. There are 8 features including median income and median house age in block group, average number of rooms per household, block group population. We standardize all covariates and the response to have mean 0 and standard deviation 1 as in Fong & Holmes (2021); Bhagwat et al. (2025). We denote $\mathbf{X} \in \mathbb{R}^9$ to be the feature vector (the first component is 1) in the California Housing Data. We give the Bayesian candidate models in CBMA and the Bayesian models in CB as follows.

We consider three different Bayesian models for CB and CBMA: i) CBMA1 includes 3 candidates: a linear model with all 8 features, a quadratic model with all 8 features, a linear model with 2 features. ii) CBMA2 includes 3 linear models, each with 2 features. iii) CBMA3 includes 2 quadratic models and one linear model, each with 2 features. iv) CB1: a linear model with all 8 features. v) CB2: a linear model with 6 features. vi) CB3: a quadratic model with 4 features.

The linear model with all 8 features:

$$Y = a_0 + \mathbf{X}^\top \mathbf{a} + \epsilon,$$

where $\mathbf{a} \sim \mathcal{N}(\mathbf{0}, 3 * \mathbf{I}_{9 \times 9})$, $\epsilon \sim \mathcal{N}(0, 1)$. The linear models with partial features have a similar form, but use less features in $\mathbf{X}$. The quadratic model with all 8 features includes all quadratics of each feature and the cross terms.

Table 3: Simulation Study: Comparison of the fitting times (in seconds) for CNB, CNB-split, CBMA1 (include the true Bayesian model), CBMA2, CBMA3, CB1 (with the true Bayesian model), CB2, CB3, CQR, LCP, CPCG with respect to different sample sizes.

| $n$ | CNB | CNB-split | CBMA1 | CBMA2 |
|---|---|---|---|---|
| 100 | $2.8 \pm 0.06$ | $0.27 \pm 0.09$ | $2.73 \pm 0.21$ | $2.97 \pm 0.70$ |
| 300 | $3.4 \pm 0.06$ | $0.53 \pm 0.11$ | $3.82 \pm 0.38$ | $4.10 \pm 0.33$ |
| 600 | $3.4 \pm 0.02$ | $1.01 \pm 0.28$ | $4.51 \pm 0.13$ | $4.11 \pm 0.57$ |
| 1000 | $3.7 \pm 0.12$ | $1.78 \pm 0.36$ | $4.78 \pm 0.22$ | $5.12 \pm 0.49$ |

| $n$ | CBMA3 | CB1 | CB2 | CB3 |
|---|---|---|---|---|
| 100 | $2.73 \pm 0.31$ | $0.89 \pm 0.07$ | $0.98 \pm 0.01$ | $0.89 \pm 0.08$ |
| 300 | $3.98 \pm 0.32$ | $1.43 \pm 0.09$ | $1.32 \pm 0.01$ | $1.27 \pm 0.04$ |
| 600 | $4.23 \pm 0.12$ | $2.40 \pm 0.02$ | $2.29 \pm 0.04$ | $2.11 \pm 0.01$ |
| 1000 | $5.09 \pm 0.07$ | $3.10 \pm 0.02$ | $2.87 \pm 0.02$ | $3.04 \pm 0.01$ |

| $n$ | CQR | LCP | CPCG |
|---|---|---|---|
| 100 | $0.35 \pm 0.05$ | $0.53 \pm 0.04$ | $0.62 \pm 0.10$ |
| 300 | $0.66 \pm 0.05$ | $0.72 \pm 0.03$ | $0.77 \pm 0.02$ |
| 600 | $0.85 \pm 0.02$ | $0.93 \pm 0.02$ | $1.00 \pm 0.09$ |
| 1000 | $0.97 \pm 0.01$ | $1.11 \pm 0.01$ | $1.34 \pm 0.01$ |

Table 4: California Dataset: Comparison of the coverage rate for CNB, CNB-split, CBMA1 (include the true Bayesian model), CBMA2, CBMA3, CB1 (with the true Bayesian model), CB2, CB3, CQR, LCP, CPCG with respect to different sample sizes.

| $n$ | CNB | CNB-split | CBMA1 | CBMA2 |
|---|---|---|---|---|
| 100 | $0.795 \pm 0.090$ | $0.797 \pm 0.050$ | $0.793 \pm 0.038$ | $0.803 \pm 0.058$ |
| 300 | $0.806 \pm 0.090$ | $0.799 \pm 0.039$ | $0.804 \pm 0.031$ | $0.795 \pm 0.040$ |
| 600 | $0.806 \pm 0.051$ | $0.798 \pm 0.039$ | $0.799 \pm 0.017$ | $0.800 \pm 0.018$ |
| 1000 | $0.801 \pm 0.031$ | $0.800 \pm 0.024$ | $0.801 \pm 0.017$ | $0.799 \pm 0.015$ |

| $n$ | CBMA3 | CB1 | CB2 | CB3 |
|---|---|---|---|---|
| 100 | $0.794 \pm 0.032$ | $0.801 \pm 0.027$ | $0.803 \pm 0.030$ | $0.790 \pm 0.028$ |
| 300 | $0.802 \pm 0.031$ | $0.800 \pm 0.022$ | $0.808 \pm 0.025$ | $0.795 \pm 0.019$ |
| 600 | $0.805 \pm 0.019$ | $0.809 \pm 0.020$ | $0.797 \pm 0.013$ | $0.798 \pm 0.015$ |
| 1000 | $0.800 \pm 0.011$ | $0.800 \pm 0.012$ | $0.799 \pm 0.014$ | $0.797 \pm 0.011$ |

| $n$ | CQR | LCP | CPCG |
|---|---|---|---|
| 100 | $0.815 \pm 0.101$ | $0.806 \pm 0.088$ | $0.799 \pm 0.092$ |
| 300 | $0.808 \pm 0.063$ | $0.799 \pm 0.055$ | $0.801 \pm 0.068$ |
| 600 | $0.803 \pm 0.065$ | $0.802 \pm 0.062$ | $0.798 \pm 0.057$ |
| 1000 | $0.801 \pm 0.037$ | $0.799 \pm 0.041$ | $0.799 \pm 0.056$ |

In Table 4, we present the coverage rates of the models with standard errors.

We report the running time for each method on real data experiments in table 5.

## D.2 OTHER REAL DATASETS

Table 5: California Dataset: Comparison of the running times for CNB, CNB-split, CBMA1 (include the true Bayesian model), CBMA2, CBMA3, CB1 (with the true Bayesian model), CB2, CB3, CQR, LCP, CPCG with respect to different sample sizes.

| $n$ | CNB | CNB-split | CBMA1 | CBMA2 |
|---|---|---|---|---|
| 100 | $3.53 \pm 0.12$ | $0.20 \pm 0.01$ | $3.23 \pm 0.21$ | $2.57 \pm 0.20$ |
| 300 | $5.37 \pm 0.62$ | $0.86 \pm 0.01$ | $5.10 \pm 0.38$ | $5.93 \pm 0.50$ |
| 600 | $7.62 \pm 0.52$ | $2.23 \pm 0.03$ | $7.76 \pm 0.28$ | $7.69 \pm 0.38$ |
| 1000 | $8.28 \pm 0.36$ | $3.04 \pm 0.01$ | $8.63 \pm 0.12$ | $8.89 \pm 0.10$ |

| $n$ | CBMA3 | CB1 | CB2 | CB3 |
|---|---|---|---|---|
| 100 | $3.38 \pm 0.21$ | $0.96 \pm 0.02$ | $0.96 \pm 0.01$ | $1.01 \pm 0.02$ |
| 300 | $5.89 \pm 0.17$ | $1.64 \pm 0.02$ | $1.32 \pm 0.03$ | $1.73 \pm 0.03$ |
| 600 | $7.80 \pm 0.10$ | $2.81 \pm 0.01$ | $2.76 \pm 0.01$ | $2.91 \pm 0.01$ |
| 1000 | $8.91 \pm 0.09$ | $4.14 \pm 0.02$ | $3.90 \pm 0.02$ | $4.52 \pm 0.01$ |

| $n$ | CQR | LCP | CPCG |
|---|---|---|---|
| 100 | $0.38 \pm 0.03$ | $0.43 \pm 0.02$ | $0.57 \pm 0.05$ |
| 300 | $0.52 \pm 0.02$ | $0.62 \pm 0.01$ | $0.69 \pm 0.03$ |
| 600 | $0.56 \pm 0.03$ | $0.67 \pm 0.01$ | $0.73 \pm 0.04$ |
| 1000 | $0.79 \pm 0.01$ | $0.86 \pm 0.01$ | $0.89 \pm 0.02$ |

Table 6: Crime Dataset: Comparison of the coverage rate for CNB, CNB-split, CBMA, CB, CQR, LCP, CPCG with respect to different sample sizes.

| $n$ | CNB | CNB-split | CBMA | CB |
|---|---|---|---|---|
| 100 | $0.801 \pm 0.080$ | $0.797 \pm 0.066$ | $0.802 \pm 0.094$ | $0.795 \pm 0.078$ |
| 300 | $0.799 \pm 0.082$ | $0.800 \pm 0.062$ | $0.801 \pm 0.088$ | $0.804 \pm 0.086$ |
| 600 | $0.805 \pm 0.078$ | $0.801 \pm 0.054$ | $0.802 \pm 0.063$ | $0.801 \pm 0.072$ |
| 1000 | $0.793 \pm 0.061$ | $0.796 \pm 0.033$ | $0.799 \pm 0.075$ | $0.801 \pm 0.050$ |

| $n$ | CQR | LCP | CPCG |
|---|---|---|---|
| 100 | $0.809 \pm 0.100$ | $0.797 \pm 0.094$ | $0.810 \pm 0.104$ |
| 300 | $0.810 \pm 0.080$ | $0.795 \pm 0.087$ | $0.808 \pm 0.096$ |
| 600 | $0.805 \pm 0.073$ | $0.796 \pm 0.058$ | $0.800 \pm 0.087$ |
| 1000 | $0.803 \pm 0.049$ | $0.802 \pm 0.060$ | $0.805 \pm 0.063$ |

The community crime dataset includes 1993 data points, the bike sharing demand dataset includes 10886 data points, and the concrete compressive strength dataset includes 1030 data points. For each of these datasets, we consider one CB model that assigns Bayesian priors for the coefficients of the regression models. We also fit one CBMA model that include three Bayesian candidate models for each of these datasets. We include the coverage rates for each of methods with respect to the sample size in Tables 6, 7, 8. Fix $\alpha = 0.2$, both methods produce prediction sets with almost 0.8 coverage rate.

## E  PROOFS OF THE MAIN RESULTS

**Proof of Theorem 1:**

**Proof:**

Table 7: Bike Dataset: Comparison of the coverage rate for CNB, CNB-split, CBMA, CB, CQR, LCP, CPCG with respect to different sample sizes.

| $n$ | CNB | CNB-split | CBMA | CB |
|---|---|---|---|---|
| 100 | $0.796 \pm 0.103$ | $0.799 \pm 0.073$ | $0.796 \pm 0.089$ | $0.803 \pm 0.058$ |
| 300 | $0.798 \pm 0.099$ | $0.802 \pm 0.073$ | $0.797 \pm 0.096$ | $0.800 \pm 0.073$ |
| 600 | $0.800 \pm 0.059$ | $0.800 \pm 0.055$ | $0.797 \pm 0.079$ | $0.800 \pm 0.080$ |
| 1000 | $0.803 \pm 0.051$ | $0.802 \pm 0.043$ | $0.800 \pm 0.068$ | $0.801 \pm 0.061$ |

| $n$ | CQR | LCP | CPCG |
|---|---|---|---|
| 100 | $0.804 \pm 0.089$ | $0.803 \pm 0.080$ | $0.807 \pm 0.102$ |
| 300 | $0.799 \pm 0.083$ | $0.800 \pm 0.067$ | $0.804 \pm 0.099$ |
| 600 | $0.801 \pm 0.075$ | $0.799 \pm 0.061$ | $0.799 \pm 0.070$ |
| 1000 | $0.801 \pm 0.050$ | $0.800 \pm 0.057$ | $0.800 \pm 0.059$ |

Table 8: Concrete Dataset: Comparison of the coverage rate for CNB, CNB-split, CBMA, CB, CQR, LCP, CPCG with respect to different sample sizes.

| $n$ | CNB | CNB-split | CBMA | CB |
|---|---|---|---|---|
| 100 | $0.800 \pm 0.071$ | $0.802 \pm 0.066$ | $0.802 \pm 0.074$ | $0.800 \pm 0.033$ |
| 300 | $0.800 \pm 0.066$ | $0.799 \pm 0.051$ | $0.798 \pm 0.065$ | $0.802 \pm 0.64$ |
| 600 | $0.798 \pm 0.071$ | $0.803 \pm 0.055$ | $0.803 \pm 0.040$ | $0.800 \pm 0.044$ |
| 1000 | $0.799 \pm 0.034$ | $0.801 \pm 0.040$ | $0.800 \pm 0.052$ | $0.800 \pm 0.038$ |

| $n$ | CQR | LCP | CPCG |
|---|---|---|---|
| 100 | $0.805 \pm 0.060$ | $0.801 \pm 0.083$ | $0.800 \pm 0.081$ |
| 300 | $0.806 \pm 0.072$ | $0.803 \pm 0.075$ | $0.801 \pm 0.069$ |
| 600 | $0.803 \pm 0.056$ | $0.799 \pm 0.076$ | $0.802 \pm 0.066$ |
| 1000 | $0.801 \pm 0.038$ | $0.800 \pm 0.069$ | $0.800 \pm 0.043$ |

The data generating process of the proposed CNB framework can be written in a hierarchical form

$$Y_i | X_i = x_i; \theta_i \sim f(y_i | x_i; \theta_i), \text{ for } i \in \{1, \cdots, n+1\}; \quad \theta_i | P \sim P; \quad P \sim \mathrm{DP}(a, H).$$

Therefore, the posterior predictive density is computed by

$$p(y_i | x_i; Z_{1:n+1}) = \mathbb{E}_{P|Z_{1:n+1}} \left[ \int f(y_i | x_i; \theta) dP(\theta) \right],$$

where the expectation is taken with respect to the posterior $P|Z_{1:n+1}$. It is trivial that the posterior $P|Z_{1:n+1}$ is invariant under permutations of $Z_{1:n+1}$. Therefore, $\mathbb{E}_{P|Z_{1:n+1}} \left[ \int f(y_i | x_i; \theta) dP(\theta) \right]$ is invariant under permutations of $Z_{1:n+1}$. This directly implies

$$\sigma_i^{\mathrm{CNB}} = p(y_i | x_i, Z_{1:n+1}) = \mathbb{E}_{P|Z_{1:n+1}} \left[ \int f(y_i | x_i; \theta) dP(\theta) \right]$$

is invariant under permutations of $Z_{1:n+1}$ and thus $\sigma_i^{\mathrm{CNB}}$ is a valid conformity score. The conformal prediction set $C_\alpha(X_{n+1})$ is constructed through the classic conformal inference procedure using $\sigma_i^{\mathrm{CNB}}$. Thus, equation 17 and equation 18 hold (Vovk et al., 2005; Lei et al., 2018).

**Proof of Theorem 2:**

**Proof:**

The result follows from Tokdar (2006); Pati et al. (2013); Ghosal et al. (1999) and Hoff (2023). Let $p^*(y|x)$ is the true conditional density of $Z_{1:n}$ on $\mathbb{R} \times \mathbb{R}^d$. Denote the true distribution of $Z_{1:n}$ as

$P^*$. Let $\Pi$ a the Bayesian nonparametric prior, for example, the DP mixture model. Then, for any dataset $Z_{1:n}$, the posterior predictive density is denoted as $p_\Pi(y|x; Z_{1:n})$.

We first state the following assumptions, under which, the DPMM posterior predictive density converges to the true posterior predictive density in probability.

A1. $\Pi$ assigns positive Kullback–Leibler support to $P^*$ and satisfies the regularity conditions ensuring posterior predictive consistency for $p^*$:

$$\sup_{(x,y)\in K} |p_\Pi(y|x; Z_{1:m}) - p^*(y|x)| \xrightarrow{P^*} 0$$

as $m \to \infty$ for every compact $K \subset \mathbb{R} \times \mathbb{R}^d$.

A2. The distribution of the oracle score $\sigma_i^*$ admits a continuous cdf with density bounded in a neighborhood of its $1-\alpha$ quantile; moreover, for $P^*$-a.e. $X_{n+1}$ the boundary $\partial C^*(X_{n+1})$ has measure 0.

All these assumptions are satisfied in our setting under the DPMM prior with choices of $f(y|x;\theta)$ as described in Section 3.4 (Tokdar, 2006; Pati et al., 2013; Ghosal et al., 1999). By A1, $p_\Pi(y|x; Z_{1:m}) \to p^*(y|x)$ uniformly on compacts in $P^*$-probability as $m \to \infty$. Taking $m = n+1$, we have

$$\sup_{(x,y)\in K} |p_\Pi(y|x; Z_{1:n+1}) - p^*(y|x)| \xrightarrow{P^*} 0$$

By the fact that $\{(X_i, Y_i)\}_{i=1}^n$ are iid and independent of $X_{n+1}$, we are able to take $K$ large enough so that all $\{(X_i, Y_i)\}_{i=1}^n \cup \{(X_{n+1}, y)\}$ lies in $K$, for any $y \in \mathcal{Y}$, the candidate set. Therefore,

$$\mathcal{I}_n := \max\left\{ \max_{1\leq i\leq n} \sup_{y\in\mathcal{Y}} |\sigma_i^{CNB} - \sigma_i^*|, \sup_{y\in\mathcal{Y}} |\sigma_{n+1}^{CNB} - \sigma_{n+1}^*| \right\} \xrightarrow{P^*} 0. \tag{25}$$

For each $y \in \mathcal{Y}$, define the CDF of the conformity scores as

$$F_n^{CNB}(t;y) := \frac{1}{n+1} \sum_{i=1}^{n+1} \mathbb{1}(\sigma_i^{CNB} \leq t)$$

$$F_n^*(t;y) := \frac{1}{n+1} \sum_{i=1}^{n+1} \mathbb{1}(\sigma_i^* \leq t)$$

where we recall $\sigma_i^{CNB} = p(Y_i|X_i, \{(X_i, Y_i)\}_{i=1}^n \cup \{(X_{n+1}, y)\})$ and $\sigma_i^*(y) = p^*(Y_i|X_i, \{(X_i, Y_i)\}_{i=1}^n \cup \{(X_{n+1}, y)\})$. Then, $r^{CNB}(y) = F_n^{CNB}(\sigma_{n+1}^{CNB}; y)$ and $r^*(y) = F_n^*(\sigma_{n+1}^*; y)$.

If $\max_{1\leq i\leq n+1} |\sigma_i^{CNB} - \sigma_i^*| \leq \epsilon$, we have

$$|F_n^{CNB}(t;y) - F_n^*(t;y)| \leq \frac{1}{n+1} \sum_{i=1}^{n+1} \mathbb{1}(|\sigma_i^* - t| \leq \epsilon).$$

By setting $t = \sigma_{n+1}^{CNB}$, we have

$$|r^{CNB}(y) - r^*(y)| \leq \frac{1}{n+1} \sum_{i=1}^{n+1} \mathbb{1}(|\sigma_i^* - \sigma_{n+1}^*| \leq 2\epsilon) \leq 2L\epsilon + o_{P^*}(1), \tag{26}$$

where we use the fact that the distribution of $\sigma_i^*$ is bounded by $L$ in a neighborhood of its $1-\alpha$ quantile. Taking $\epsilon = \Delta_n$, we have

$$\sup_{y\in\mathcal{Y}} |r^{CNB}(y) - r^*(y)| \xrightarrow{P^*} 0.$$

This implies that the indicator functions $\mathbb{1}(r^{CNB}(y) > \alpha)$ and $\mathbb{1}(r^*(y) > \alpha)$ differ only on the band $\{y : |r^*(y) - \alpha| \leq \eta_n\}$ with some $\eta_n \to 0$ in probability uniformly on $\mathcal{Y}$. By assumption A2, we immediately have (Hoff, 2023)

$$\mathbb{E}[\mathcal{V}(C_\alpha^{CNB}(X_{n+1}) - \mathcal{V}(C_\alpha^*(X_{n+1})] \xrightarrow{P^*} 0,$$

This completes the proof.