# OpenReview forum: "CNB: A Bayesian Nonparametric Approach to Optimal Conformal Prediction"
_ICLR.cc/2026/Conference — Submitted to ICLR 2026_

### Official Review · Reviewer_d3HW · 2025-10-28

**Soundness:** 2
**Presentation:** 2
**Contribution:** 2
**Rating:** 2
**Confidence:** 4

**Summary:**

The paper studies constructing the optimal conformal prediction set under Bayesian modeling.
In particular, it proposes using a non-paramtric modeling approach to approximate the
true Bayesian model, thereby approximating the optimal prediction set under the (unknown) true Bayesian model.
The proposed method has the frequentist guarantee of conformal inference, and asymptotic Bayesian optimality under
certain assumptions. The proposed method is evaluated on synthetic and real dataset.

**Strengths:**

1. The proposed method appears to combine the benefit of Bayesian modeling with the distribution-free guarantee
offered by conformal inference.
2. The empirical results demonstrate the preliminary benefits of the proposed approach.

**Weaknesses:**

1. The exposition of the methods and theories lacks sufficient rigor (to be specified in the question section),
making it hard to properly evaluate/interpret the results.
2.The numerical comparisons are limited, and the reported advantages are not particularly convincing—for example, the proposed prediction interval is substantially shorter than those of competing methods only when the sample size is very large.
3. The methodological innovation seems to be limited.

**Questions:**

1. On page 5, equation (11), I am confused by the use of the running variable $i$ on the right-hand side, and by whether it is correct that it sums over $(x_i,y_i)$ for $i \in\{1,2,\ldots\}$.
This is confusing since (1) $(x_i,y_i)$ is not defined for $i>n$ and (2) in the case of truncation with $K_n<n$, then obviously $\sigma_i$ is no longer invariant to the permutation of $Z_{i=1}^{n+1}$.
2. Theorem 1 claims that the proposed method achieves finite-sample coverage. If I understand correctly, the result only applies to the hypothetical set by enumerating all the
values of $y \in \mathbb{R}$; what guarantees are provided for the approximation via the AOI strategy, which is also the actual computable prediction set used in the numerical experiments?
3. In Theorem 2, what does "optimality" refer to exactly? Is it the optimal prediction set with some full Bayesian model? Also, I wonder how the convergence depends on the modeling choice
of $f(y \mid x;\theta)$ and the dimension of $X$.

---

> ### Author Response · Authors · 2025-11-19
> **Rebuttal by Authors**
>
> We thank the reviewer for their careful reviewing and feedback.
>
> We have revised the manuscript according to your valuable feedback. **Please review the updated submission** and let us know if you have further concerns and questions.
>
> > Weakness 1
>
> We will answer your specific questions in later responses. We hope these responses can help to evaluate our work.
>
> To better clarify the proposed CNB method and emphasize its benefits, we would like to give *a short summary*. Given a fixed coverage rate $1-\alpha$, we aim to find the optimal conformal prediction set, which is the one with the minimal expected interval length. Under the fixed coverage rate, such optimal prediction set is the most informative one.  It is known (shown by Hoff (2023)) that the Bayesian conformal prediction set **under a correctly specified Bayesian model** is optimal across all conformal prediction sets. However, in practice, a practitioner rarely knows “the correctly specified Bayesian model”, therefore, the standard conformal Bayesian prediction set may not be optimal if the Bayesian model is misspecified. We address this challenge by proposing CNB that gives the optimal prediction set without the necessary of specifying a correct Bayesian model. We theoretically show the CNB prediction set is valid and converges to the optimal prediction set.
>
> > Weakness 2
>
> To emphasize the performance of the proposed CNB method, we have further run experiments on three more real datasets from UCI: community crime dataset, bike sharing demand dataset, and concrete compressive strength dataset in Section 5.
>
> In addition to the comparisons with two Bayesian conformal methods: conformal Bayesian prediction (CB) and CBMA, we add three more non-Bayesian conformal methods including Conformalized quantile regression (CQR) (Romano et al., 2019), Localized conformal prediction (LCP) (Guan, 2023), Conformal prediction with conditional guarantees (CPCG) (Gibbs et al., 2025) for each dataset in our experiments.  We have re-organized the experiment section and added interpretations with respect to these experiments. From those experiments, the proposed CNB method always give the optimal prediction sets comparing with other methods. In particular, CNB method consistently produces **the optimal (narrowest) prediction sets for small sample size** on the simulation dataset, the California housing dataset,  the bike sharing demand dataset, and the concrete compressive strength dataset.
>
> > Weakness 3
>
> We agree with the reviewer on that there are experiment limitations and some clarification issues. We are trying our best to add more experiments and provide better clarifications according to your valuable feedback. However, we respectfully disagree that the methodological contribution is limited. Constructing optimal conformal prediction sets at a fixed coverage level is a well recognized challenge (e.g., Hoff, 2023; Bhagwat et al., 2025; Candès, Lei, & Ren, 2023). Existing approaches, including conformal Bayesian inference (CB; Hoff, 2023) and conformal Bayesian inference under model averaging (CBMA; Bhagwat et al., 2025), address this challenge assuming the correct Bayesian model is known or correctly specified, which is an assumption that is rarely feasible in practice. Our CNB method tackles the same problem without requiring prior knowledge of the correct model, by leveraging a Bayesian nonparametric predictive within the conformal framework. The effectiveness of CNB is supported by theoretical guarantees and empirical evidence on simulated data and four real datasets. We hope the revised manuscript addresses the reviewer’s concerns.
>
> > Question 1
>
> The index $i$ for $w$ and $\theta$ in equation (11) is a typo. The index for $w$ and $\theta$ should be $j$ and the sums of $i$ should be sums of $j$. We have corrected this mistake in the updated version.
>
> In Section 3.2, the computation of full CNB prediction set, we truncate the infinite sum in (11) at some $K_n$ such that $1-\sum_{j=1}^{K_n}w_j^{(n+1)} \leq \epsilon$. This truncation does not break the exchangeability of the corresponding conformity scores. Reasons: The truncation $K_n$ is a deterministic function of the posterior weights $\{w_j^{(n+1)}\}$, hence a symmetric functional of the set $Z_{1:n+1}$. The posterior sequence $\{(w_j^{(n+1)},\theta_j^{(n+1)})\}$ is unchanged by permuting $Z_{1:n+1}$. Therefore, the truncation index $K_n$ remains the same before and after the permutation. The truncation $K_n$ does not break the exchangeability of the corresponding conformity scores.  We have added this discussion in Appendix B.1 as a clarification.

---

> ### Author Response · Authors · 2025-11-19
> **Rebuttal by Authors**
>
> > Question 2
>
> Your observation about Theorem 1 is correct. Theorem 1 holds for $C_{\alpha}^{CNB}(X_{n+1})$, which is constructed through a full conformal procedure. It is possible to compute $C_{\alpha}^{CNB}(X_{n+1})$ by enumerating all the values of $y\in \mathcal{Y}$.
>
> However, this computation requires retrain the model for each $y$, which is computational expensive. We use AOI as a way to avoid retraining the model for each $y$. This is a commonly used technique in conformal Bayesian inference, e.g., Fong & Holmes (2021), Hoff (2023), Bhagwat et al., (2025).
>
> The AOI strategy produces an approximation $\hat{C}\_{\alpha}^{CNB}(X_{n+1})$ of $C_{\alpha}^{CNB}(X_{n+1})$. The approximation error depends on the quantity and quality of MCMC draws, with more sufficient MCMC draws, the approximation error is smaller. That said, the AOI procedure is used to reduce computation. The coverage rates of the experiments (Appendix C and D) show that this AOI procedure does not significantly affect coverage.
>
>  We agree that there should be a carefully discussion of using this AOI approximation. To formally quantify the effect of the AOI approximation to Theorem 1, we have added one remark (Remark 6 in Appendix B.1) as follows.
>
> Assume that there exists $\epsilon>0$ and $\delta \in [0,1)$ such that
> $\mathbb{P}(\sup\_{y\in\mathcal{Y}} \max\_{1\leq i\leq n+1} |\hat{\sigma}\_i^{CNB}-\sigma\_i^{CNB}|\leq \epsilon )\geq 1-\delta$.
> We further assume the density of $\sigma\_{n+1}^{CNB}$ is bounded by $L$. Then, we have
>
>    $ \mathbb{P}(Y\_{n+1}\in \hat{C}\_{\alpha}^{CNB}(X\_{n+1}) )\geq 1-\alpha-\delta-4L\epsilon$.
>
> The formal proof of this remark is given in Appendix B.1
>
> > Question 3
>
> Optimality means that, for a fixed coverage level, the conformal prediction set with the smallest expected length is optimal. For any prediction set $C$, $\mathcal{V}(C)$ represents the length of the prediction set. Shown by Hoff (2023), $\hat{C}\_{\alpha}^*(X\_{n+1})$  (constructed by conformal Bayesian under the correct Bayesian model) is the optimal prediction set.
>
> Theorem 2 shows the expected length of the proposed CNB prediction set converges to the expected length of the optimal prediction set $\hat{C}\_{\alpha}^*(X\_{n+1})$.
>
> We include multiple choices of $f(y|x)$ in Section 3.4, both works for high dimensional $x $. As long as the assumptions of $f(y|x)$  given under Theorem 2 and in the proof of Theorem 2 are satisfied, Theorem 2 holds. In particular, these assumptions hold for DPMM joint models and regression models, which are the ones used in the experiments.
>
> The convergence rate in Theorem 2 inherits the rate of posterior predictive convergence of the Bayesian nonparametric model, which in turn depends on the dimension of $X$. In high dimension case, standard DPMM convergence rates deteriorate  (Tokdar, 2006; Pati et al., 2013; Ghosal et al., 1999), and the convergence rate in Theorem 2 is slower.
>
> *Peter Hoff. Bayes-optimal prediction with frequentist coverage control. Bernoulli,  2023.*
>
> *Pankaj Bhagwat, Linglong Kong, and Bei Jiang. Cbma: Improving conformal prediction through Bayesian model averaging. In The Thirteenth International Conference on Learning Representations, 2025*
>
> *Isaac Gibbs, John J Cherian, and Emmanuel J Candes. Conformal prediction with conditional guarantees. Journal of the Royal Statistical Society Series B: Statistical Methodology, 2025.*
>
> *Leying Guan. Localized conformal prediction: A generalized inference framework for conformal prediction. Biometrika, 2023.*
>
> *Yaniv Romano, Evan Patterson, and Emmanuel Candes. Conformalized quantile regression. Advances in neural information processing systems, 32, 2019.*
>
> *Subhashis Ghosal, Jayanta K Ghosh, and RV Ramamoorthi. Posterior consistency of Dirichlet mixtures in density estimation. The Annals of Statistics, 27(1):143–158, 1999.*
>
> *Debdeep Pati, David B Dunson, and Surya T Tokdar. Posterior consistency in conditional distribution estimation. Journal of multivariate analysis, 116:456–472, 2013.*
>
> *Surya T Tokdar. Posterior consistency of Dirichlet location-scale mixture of normals in density estimation and regression. Sankhya: The Indian Journal of Statistics, pp. 90–110, 2006.*

---

### Official Review · Reviewer_uUhA · 2025-11-01

**Soundness:** 3
**Presentation:** 3
**Contribution:** 3
**Rating:** 6
**Confidence:** 3

**Summary:**

The paper proposes Conformalized Nonparametric Bayes (CNB), a framework that combines Bayesian predictive modeling with distribution-free conformal calibration to construct prediction sets that remain valid and efficient even under model misspecification (addressing a key limitation of existing conformal prediction methods). CNB leverages Dirichlet Process Mixtures Models (DPMM) to model conditional densities and define conformity scores that account for model uncertainty. Theoretical results guarantee finite-sample validity and show convergence to optimal efficiency, while empirical results on synthetic and real data demonstrate that CNB produces smaller, more adaptive prediction sets that maintain target coverage, whereas other methods perform poorly when the underlying Bayesian model is mis specified.

**Strengths:**

The paper addresses a limitation of conformal prediction, inefficiency under model misspecification, by integrating it with Bayesian nonparametric inference. The formulation is principled, well-motivated, and clearly presented.

Theoretical results guarantee finite-sample validity and asymptotic optimal efficiency, showing that CNB preserves the coverage properties of conformal prediction while converging to the smallest valid prediction sets under the true data-generating model.

The use of DPMMs allows the method to capture complex, multimodal conditional densities, providing adaptability beyond parametric Bayesian models and accounting for model misspecification.

The framework seems to be agnostic to the exact Bayesian model used, it can integrate different predictive models such as DPMMs with Gaussians or DPMMs with regressions.

Experimental results on synthetic and real datasets demonstrate that CNB consistently produces competitive and efficient prediction intervals across a range of sample sizes, empirically supporting its theoretical claims.

**Weaknesses:**

1. The experimental evaluation is relatively narrow, relying mainly on small synthetic settings and a single real dataset.

2. Experiments are conducted at a fixed error rate of 0.2, without discussion of why this value was chosen or how performance might vary with different target coverages.

3. It would be valuable to include experiments or discuss the integration of CNB with pretrained or foundation models (e.g., tabular or time-series models) to show how the framework performs in large-scale or representation-rich settings.

4. The paper does not include comparisons or discussion of adaptive conformal prediction approaches designed to handle heteroscedastic or distribution-shifted uncertainty, such as:

   •Romano, Patterson, and Candès (2019), Conformalized Quantile Regression;

   •Gibbs and Candès (2021), Adaptive Conformal Inference Under Distribution Shift;

   •Gibbs, Cherian, and Candès (2024), Conformal Prediction with Conditional Guarantees;

   •Guan (2023), Localized Conformal Prediction;

   •Amoukou and Brunel (2023), Adaptive Conformal Prediction by Reweighting Nonconformity Scores.

Including comparisons or at least a discussion relative to these adaptive methods would help clarify CNB’s practical advantages and positioning within the broader conformal prediction landscape.

**Questions:**

See weaknessess

---

> ### Author Response · Authors · 2025-11-19
> **Rebuttal by Authors**
>
> We thank the reviewer for their careful reviewing and feedback.
>
> We have revised the manuscript according to your valuable feedback. **Please review the updated submission** and let us know if you have further concerns and questions.
>
> > Weakness 1
>
> We have further run experiments on three more real datasets from UCI: community crime dataset, bike sharing demand dataset, and concrete compressive strength dataset in Section 5. Extra results of these experiments are included in Appendix D.
>
> > Weakness 2
>
> We follow the standard conformal inference setting, we construct a prediction set at a given coverage level. We fix the miscoverage rate at $\alpha=0.2$, so the prediction sets are expected to cover the true outcome with at least 0.8 probability. $\alpha$ is predetermined by users. Give a set of data and a fixed model, the length of the conformal prediction set is increasing as $\alpha$ decreases. This setting is common in the literature, for example  $\alpha=0.1$ is fixed in Romano, Patterson, and Candès (2019), Gibbs and Candès (2021), Gibbs, Cherian, and Candès (2024), Amoukou and Brunel (2023).
>
> > Weakness 3
>
> We appreciate that you mention a potential application of the proposed CNB method. In Appendix B.2, we have added a formal discussion of how pretrained model can be applied in the CNB framework. This work mainly focuses on the introduction of the CNB method and the theoretical guarantee. Therefore, we would keep focusing on the current study scope and keep the application of the CNB method as one of the future research directions. Please find the discussion as follows.
>
> CNB is model agnostic and composes naturally with learned representations. Let $g: \mathbb{R}^d \to \mathbb{R}^s$ be any upstream model (e.g., a tabular or time series backbone) trained on the training split only, and define an embedding $\phi(x)=g(x)$. We may (i) run representation based CNB, replacing $x$ by $\phi(x)$ and fitting the BNP conditional $p(y|\phi(x))$(e.g., a DPMM for $(\phi(x),y))$, or (ii) run a residual CNB, using a backbone baseline $\mu_g(x)$ (and optionally $\sigma_g^2(x)$) and modeling the residual $r=y-\mu_g(x)$ with a BNP mixture so that the predictive for $y|x$ is the convolution of the baseline and the BNP residual model.
>
> Both cases only change the predictive distributions used in the computation of the CNB scores and leave the conformal machinery untouched. Since $\phi$ (and $\mu_g(x)$) are fixed prior to calibration, exchangeability is preserved and the finite sample coverage guarantee of Theorem 1 remains valid. In practice, using $\phi(x)$ reduces effective dimension and let the BNP approach discover local regimes in representation space, often yielding shorter sets at the same coverage. This makes CNB a representation compatible conformal method, practitioners can keep their preferred pretrained model and simply use the outputs into the CNB predictive.
>
>
> > Weakness 4
>
> In addition to the comparisons with two Bayesian conformal methods: conformal Bayesian prediction (CB) and CBMA, we add three more non-Bayesian conformal methods including Conformalized quantile regression (CQR) (Romano et al., 2019), Localized conformal prediction (LCP) (Guan, 2023), Conformal prediction with conditional guarantees (CPCG) (Gibbs et al., 2025) for each dataset in our experiments. We have re-organized the experiment section and added interpretations with respect to these experiments. From those experiments, the proposed CNB method always give the optimal prediction sets comparing with other methods.
>
> The methods in *Gibbs and Candès (2021), Adaptive Conformal Inference Under Distribution Shift* and *Amoukou and Brunel (2023), Adaptive Conformal Prediction by Reweighting Nonconformity Scores* are mainly for constructing conformal prediction sets under distribution shift. Therefore, they do not align with the main theme of this work. We will keep them as references in the related work section.
>
> *Isaac Gibbs, John J Cherian, and Emmanuel J Candes. Conformal prediction with conditional guarantees. Journal of the Royal Statistical Society Series B: Statistical Methodology, 2025.*
>
> *Leying Guan. Localized conformal prediction: A generalized inference framework for conformal prediction. Biometrika, 2023.*
>
> *Yaniv Romano, Evan Patterson, and Emmanuel Candes. Conformalized quantile regression. Advances in neural information processing systems, 32, 2019.*

---

### Official Review · Reviewer_hBDL · 2025-11-04

**Soundness:** 1
**Presentation:** 1
**Contribution:** 2
**Rating:** 2
**Confidence:** 3

**Summary:**

The present paper considers the application of Dirichlet Process Mixture Models in Bayesian inference coupled with conformal prediction. The authors derive the corresponding conference score, prove marginal finite-sample validity of the method, and draft the proof of its asymptotic optimality. Some experiments on synthetic and real-world data are provided.

**Strengths:**

- Conformalization of Bayesian methods is an interesting and currently underexplored area of research.
- Potentially, flexible nonparametric procedures can be a powerful tool for inference in real-world applications.

**Weaknesses:**

- The authors don't clearly convey the benefits of the proposed method compared to more standard Bayesian conformal prediction approaches. The resulting score is still a combination of function values over some samples from the posterior. Even if formally posterior is defined in a different way, one needs to justify possible benefits for practical applications. Some examples could help.

- The theoretical part doesn't contain full proof of the results. In particular, proof of Theorem 2 is not more than a sketch.

- The experimental part is extremely lightweight, with only one real-world dataset considered. One would need more detailed experimental study to comprehensively show the usefullness of the method.

**Questions:**

1. What does $\mathcal{V}$ mean in Theorem 2? This notation seems not to have been introduced.

2. What is the use of Theorem 3? Isn't it a standard result in Bayesina nonparametrics not directly related to conformal inference?

---

> ### Author Response · Authors · 2025-11-19
> **Rebuttal by Authors**
>
> We thank the reviewer for their careful reviewing and feedback.
>
> We have revised the manuscript according to your valuable feedback. **Please review the updated submission** and let us know if you have further concerns and questions.
>
> We first give **a short summary** of our work: Given a fixed coverage rate $1-\alpha$, we aim to find the optimal conformal prediction set, which is the one with the minimal expected interval length. Under the fixed coverage rate, such optimal prediction set is the most informative one.  It is known (shown by Hoff (2023)) that the Bayesian conformal prediction set **under a correctly specified Bayesian model** is optimal across all conformal prediction sets. However, in practice, a practitioner rarely knows “the correctly specified Bayesian model”, therefore, the standard conformal Bayesian prediction set may not be optimal if the Bayesian model is misspecified. We address this challenge by proposing CNB that gives the optimal prediction set without the necessary of specifying a correct Bayesian model. We theoretically show the CNB prediction set is valid and converges to the optimal prediction set.
>
> > Weakness 1
>
> **Benefits and comparisons:**
>
> (i) To construct the optimal prediction, the standard Bayesian conformal (CB) prediction needs the “correct” Bayesian model, while the proposed CNB prediction does not need to specify the Bayesian model. This benefit is illustrated in the simulation experiment: CB2, CB3 are the Bayesian conformal prediction models with misspecified Bayesian models, the prediction sets can be very wide. In contrast, CNB prediction sets are the narrowest give the same coverage rate.
>
> (ii) Your observation about the conformity score of CNB is a weighted combination of function values is correct. However, the source of those weights is different: in CB they come from a fixed finite dimensional posterior; in CNB they come from a posterior over an infinite mixture that (a) adapts the endogenous clustering structure of data, and (b) can converge to the true conditional law. Therefore, the CNB conformity score can converge to the conformity score of the Bayesian conformal inference with a correctly specified Bayesian model. This is exactly the foundation of the theoretical guarantee of using the CNB conformity score.
>
> > Weakness 2
>
> We have revised the proof of Theorem 2 in Appendix E in the updated manuscript. The theorem builds on Tokdar (2006) and Hoff (2023), and the revised proof includes the assumptions, and the details of how these results apply to the proposed CNB procedure.
>
> > Weakness 3
>
> We have further run experiments on three more real datasets from UCI: community crime dataset, bike sharing demand dataset, and concrete compressive strength dataset in Section 5.
>
> In addition to the comparisons with two Bayesian conformal methods: conformal Bayesian prediction (CB) and CBMA, we add three more non-Bayesian conformal methods including Conformalized quantile regression (CQR) (Romano et al., 2019), Localized conformal prediction (LCP) (Guan, 2023), Conformal prediction with conditional guarantees (CPCG) (Gibbs et al., 2025) for each dataset in our experiments.  We have re-organized the experiment section and added interpretations with respect to these experiments. From those experiments, the proposed CNB method always give the optimal prediction sets comparing with other methods.
>
> > Question 1
>
> For any prediction set $C$, $\mathcal{V}(C)$ represents the length of the prediction set. We have added the introduction of this notation in Theorem 2 in the updated version.
>
> > Question 2
>
> In this work, we claim the benefit of the proposed CNB method that it “captures endogenous clustering” and “model complexity adapts to data.” That claim needs at least a pointer to the actual distribution of the number of clusters under our specific Bayesian nonparametric model. Theorem 3 does exactly that, and it does it in the same hierarchy that is used to compute the conformity score. So it is the clustering behavior of the very model whose posterior predictive we feed into conformal.
>
> It also explains why CNB can help where standard conformal Bayesian inference cannot. Equation (21) characterizes the model complexity, it shows the model complexity adapts data and grows slowly with data at a rate $\log n$. This is the mechanism by which CNB can locally tailor predictive densities for different latent subgroups.
>
> *Peter Hoff. Bayes-optimal prediction with frequentist coverage control. Bernoulli,  2023.*
>
> *Isaac Gibbs, John J Cherian, and Emmanuel J Candes. Conformal prediction with conditional guarantees. Journal of the Royal Statistical Society Series B: Statistical Methodology, 2025.*
>
> *Leying Guan. Localized conformal prediction: A generalized inference framework for conformal prediction. Biometrika, 2023.*
>
> *Yaniv Romano, Evan Patterson, and Emmanuel Candes. Conformalized quantile regression. Advances in neural information processing systems, 32, 2019.*

---

### Author Response · Authors · 2025-11-29
**Summary of the paper**

We thank the area chairs for their time and effort in reviewing our work. We provide a brief summary of the paper and its contributions to help the review.

> **Summary and contributions:**

Given a fixed coverage rate $1-\alpha$, we aim to find the optimal conformal prediction set, which is the one with the minimal expected interval length. Under the fixed coverage rate, such optimal prediction set is the most informative one. It is known (shown by Hoff (2023)) that the **Bayesian conformal prediction set under a correctly specified Bayesian model is optimal** across all conformal prediction sets. However, in practice, a practitioner rarely knows “the correctly specified Bayesian model”, therefore, the standard conformal Bayesian prediction set may not be optimal if the Bayesian model is misspecified. We address this challenge by proposing Conformal Nonparametric Bayesian (CNB) prediction method that gives the optimal prediction set without the necessary of specifying a correct Bayesian model. We theoretically show the CNB prediction set is valid and converges to the optimal prediction set.

The main contributions of our work are:

(i) We introduce CNB, a conformal inference procedure with distribution free validity and theoretical guarantees that its prediction sets are asymptotically optimal at a fixed coverage level.

(ii)  We incorporate a Bayesian nonparametric mixture into CNB, which captures endogenous clustering and covariate dependent heterogeneity, yielding more expressive and robust prediction sets.

(iii) Computationally, we use the Add-One-In (AOI) importance sampling strategy to avoid refitting the model for every candidate outcome $y$ in the hypothesis set $\mathcal{Y}$. We also provide a theoretical justification of AOI to validate the conformal prediction sets using AOI within our framework.

(iv) The model complexity is fully data-driven, adapting to the sample via the Bayesian nonparametric prior, this behavior is formally characterized in Theorem 3.

---

### Author Response · Authors · 2025-11-29
**Summary of the reviewer's main concerns and our revisions**

We summarize the reviewers’ main concerns and our corresponding revisions as follows.


> **Concerns from the reviewers and our revisions:**

**Concerns from reviewers:** (i) Limited experiments. (ii) The proof of Theorem 2 does not include details. (iii) Validation of the CNB conformal inference using truncation $K_n$ and using AOI. (iv) Discussions of different $f(y|x)$ and high dimensional $X$.

**Revisions:**

(i) We have further run experiments on three more real datasets from UCI: community crime dataset, bike sharing demand dataset, and concrete compressive strength dataset in Section 5.

In addition to the comparisons with two Bayesian conformal methods: conformal Bayesian prediction (CB) and CBMA, we add three more non-Bayesian conformal methods including Conformalized quantile regression (CQR) (Romano et al., 2019), Localized conformal prediction (LCP) (Guan, 2023), Conformal prediction with conditional guarantees (CPCG) (Gibbs et al., 2025) for each dataset in our experiments. We have re-organized the experiment section and added interpretations with respect to these experiments. From those experiments, the proposed CNB method always give the optimal prediction sets comparing with other methods.

(ii) We have revised the proof of Theorem 2 in Appendix E in the updated manuscript. The theorem builds on Tokdar (2006) and Hoff (2023), and the revised proof includes the assumptions, and the details of how these results apply to the proposed CNB procedure.

(iii)  In Section 3.2, the computation of full CNB prediction set, we truncate the infinite sum in (11) at some $K_n$ such that $1-\sum_{j=1}^{K_n}w_j^{(n+1)} \leq \epsilon$. This truncation does not break the exchangeability of the corresponding conformity scores. Reasons: The truncation $K_n$ is a deterministic function of the posterior weights $\{w_j^{(n+1)}\}$, hence a symmetric functional of the set $Z_{1:n+1}$. The posterior sequence $\{(w_j^{(n+1)},\theta_j^{(n+1)})\}$ is unchanged by permuting $Z_{1:n+1}$. Therefore, the truncation index $K_n$ remains the same before and after the permutation. The truncation $K_n$ does not break the exchangeability of the corresponding conformity scores.  We have added this discussion in Appendix B.1 as a clarification.

The AOI strategy produces an approximation $\hat{C}\_{\alpha}^{CNB}(X_{n+1})$ of $C_{\alpha}^{CNB}(X_{n+1})$. The approximation error depends on the quantity and quality of MCMC draws, with more sufficient MCMC draws, the approximation error is smaller. That said, the AOI procedure is used to reduce computation. The coverage rates of the experiments (Appendix C and D) show that this AOI procedure does not significantly affect coverage.

To formally quantify the effect of the AOI approximation to Theorem 1, we have added one remark (Remark 6 in Appendix B.1) as follows.

Assume that there exists $\epsilon>0$ and $\delta \in [0,1)$ such that
$\mathbb{P}(\sup\_{y\in\mathcal{Y}} \max\_{1\leq i\leq n+1} |\hat{\sigma}\_i^{CNB}-\sigma\_i^{CNB}|\leq \epsilon )\geq 1-\delta$.
We further assume the density of $\sigma\_{n+1}^{CNB}$ is bounded by $L$. Then, we have

   $ \mathbb{P}(Y\_{n+1}\in \hat{C}\_{\alpha}^{CNB}(X\_{n+1}) )\geq 1-\alpha-\delta-4L\epsilon$.

The formal proof of this remark is given in Appendix B.1

(iv) We include multiple choices of $f(y|x)$ in Section 3.4, both works for high dimensional $x$. As long as the assumptions of $f(y|x)$  given under Theorem 2 (or in the proof of Theorem 2) are satisfied, Theorem 2 holds.

The convergence rate in Theorem 2 inherits the rate of posterior predictive convergence of the Bayesian nonparametric model, which in turn depends on the dimension of $X$. In high dimension case, standard DPMM convergence rates deteriorate  (Tokdar, 2006; Pati et al., 2013; Ghosal et al., 1999), and the convergence rate in Theorem 2 is slower.

---

### Meta-Review · Area_Chair_raon · 2026-01-08

**Summary:**

The reviewers raised several key concerns:
1. Insufficient rigor of exposition and theory (d3HW, hBDL), including the validity of truncation, guarantees of the AOI strategy, and lack of clarity about optimality.
2. Narrow experimental evaluation (uUhA, d3HW, hBDL)
3. Limited methodological innovation (d3HW)
4. Unclear benefit over standard Bayesian conformal prediction approaches (hBDL)
5. Missing comparison to adaptive prediction approaches (uUhA)

**Reviewer Concerns:**

The authors addressed the reviewer concerns as follows:
1. The authors revised the proof of Theorem 2, added an explanation that truncation does not break exchangeability, and added a theoretical statement of the effect of the AOI approximation.
2. The authors added experiments on an additional three datasets from UCI. In addition to a plot showing the average interval width across sample size, the results include a comparison of running time and coverage rate.
3. The authors clarified that unlike standard Bayesian conformal prediction approaches, which assume the presence of a correctly specified Bayesian model, the proposed method adapts to incorrectly specified models.
4. The authors clarified that the proposed method can adapt to misspecified models and therefore goes beyond previous approaches where the model must be correctly specified.
5. The authors added comparisons to three non-Bayesian conformal prediction methods (CQR, LCP, and CPCG)

Concerns 3 and 4 remain only partially resolved. Although the authors have clarified that the proposed method is more flexible, it still falls into the more general framework of Bayesian conformal prediction methods, albeit with a different choice of prior. It is unclear whether reviewers d3HW and hBDL would be persuaded by the arguments presented during the rebuttal phase. A more systematic study that clearly and convincingly demonstrates the empirical benefits of the proposed method would help to resolve these concerns.

**Reviewer Scores:**

- hBDL is about as likely as not to increase their score. The authors clarified that the proposed method is more flexible than standard Bayesian conformal prediction methods and can coverge to conformity score of a correctly specified Bayesian model. They also added a proof of Theorem 2 and added experiments on three additional datasets and three baselines. It is about equally likely that hBDL would increase their score (due to the additional experiments and proof of Theorem 2) as they would keep their score (due to the lack of conclusive empirical evidence that the proposed method outperforms standard Bayesian conformal methods empirical benefits of the proposed method).
- uUhA is very likely to keep their score. The initial review was already positive. During the rebuttal phase, the authors added experiments on three additional datasets and results using three non-Bayesian conformal prediction methods.
- d3HW is about as likely as not to increase their score. They raised concerns about insufficient rigor of method exposition & theory, limited numerical comparisons, and limited methodological innovation. It is about equally likely that they increase their score (due to the additional experiments and new theoretical results in Appendix B.1) as they would keep their score (due to the limited methodological innovation).

---

### Decision · Program_Chairs · 2026-01-26

Reject